



# Statistical validation of Aeolus L2A particle backscatter coefficient retrievals over ACTRIS/EARLINET stations in the Iberian Peninsula

Jesús Abril-Gago[1,2], Juan Luis Guerrero-Rascado[1,2], Maria João Costa[3,4], Juan Antonio Bravo-Aranda[1,2], Michaël Sicard[5,6], Diego Bermejo-Pantaleón[1,2], Daniele Bortoli[3,4,7], María José Granados-Muñoz[1,2], Alejandro Rodríguez-Gómez[5], Constantino Muñoz-Porcar[5], Adolfo Comerón[5], Pablo Ortiz-Amezcua[1,2,8], Vanda Salgueiro[3,4], Marta María Jiménez-Martín[1,2], Lucas Alados-Arboledas[1, 2]

[1] Andalusian Institute for Earth System Research (IISTA-CEAMA), Granada, 18006, Spain

[2] Applied Physics Department, University of Granada, Granada, 18071, Spain

[3] Earth Remote Sensing Laboratory (EaRSLab), University of Évora, Évora, 7000-671, Portugal

[4] Institute of Earth Sciences (ITC) and Department of Physics, University of Évora, Évora, 7000-671, Portugal

[5] CommSensLab, Department of Signal Theory and Communications, Universitat Politècnica de Catalunya (UPC), Barcelona, 08034, Spain

[6] Ciències i Tecnologies de l'Espai-Centre de Recerca de l'Aeronàutica i de l'Espai, Institut d'Estudis Espacials de Catalunya (CTE-CRAE/IEEC), Universitat Politècnica de Catalunya (UPC), Barcelona, 08034, Spain

[7] Institute of Atmospheric Sciences and Climate (ISAC-CNR), Bologna, 40129, Italy

[8] Faculty of Physics, University of Warsaw, Warsaw, 02-093, Poland

*Correspondence to*: Juan Luis Guerrero-Rascado (rascado@ugr.es)

**Abstract.** Global Observing Systems (GOS) encounter some limitations due to a lack of worldwide real-time wind measurements. In this context, the European Space Agency (ESA) has developed the Aeolus satellite mission, based on the ALADIN (Atmospheric Laser Doppler Instrument) Doppler wind lidar, aimed to obtain near real-time wind retrievals at global scale. As spin-off products, the instrument retrieves aerosol optical properties such as particle backscatter and extinction coefficients. In this work, a validation of Aeolus reprocessed (baseline 10) co-polar backscatter coefficients ($\beta_{Aeolus}^{part}$) is presented through an intercomparison with analogous ground-based measurements taken at the ACTRIS/EARLINET stations of Granada (Spain), Évora (Portugal) and Barcelona (Spain) over the period from July 2019 until October 2020. Case studies are first presented, followed by a statistical analysis. The stations are located in a hot spot between Africa and the rest of Europe, which guarantees a variety of aerosol types, from mineral dust layers to continental/anthropogenic aerosol, and allow us to test Aeolus performance under different scenarios. The so called Aeolus-like profiles ($\beta_{Aeolus\,like,355}^{part}$) are obtained from total particle backscatter coefficient and linear particle depolarization ratio ($\delta_{linear}^{part}$) profiles at 355 nm and 532 nm measured from surface, through a thorough bibliographic review of dual-polarization measurements for relevant aerosol types. Finally, the study proposes a relation for the spectral conversion of $\delta_{linear}^{part}$, which is implemented in the Aeolus-like profile calculation. The statistical results show the ability of the satellite to detect and characterize significant aerosol layers under cloud free conditions, along with the surface effect on the lowermost



measurements, which causes the satellite to largely overestimate co-polar backscatter coefficients. Finally, Aeolus standard
correct algorithm middle bin (SCAmb) shows a better agreement with ground-based measurements than the standard correct
algorithm (SCA), which tends to retrieve negative and meaningless coefficients in the clear troposphere. The implementation
of Aeolus quality flags entails a vast reduction in the number of measurements available for comparison, which affects the
statistical significance of the results, without improving significantly the statistical agreement between satellite and ground-
based measurements.

## 1 Introduction

Aerosol particles are a key component in the climate system, scattering and absorbing the solar and thermal radiation. As
highlighted by the Intergovernmental Panel on Climate Change (IPCC), the uncertainty of the radiative effects of some
aerosol components, such as black carbon (strong positive radiative forcing), organic carbon (strong negative forcing) or
mineral dust (small but yet significant negative forcing) is still exceptionally large (Myhre et al., 2013). Moreover, the
current uncertainties due to aerosol-cloud interactions do not allow an accurate assessment of their radiative forcing. Thus,
more comprehensive datasets of the properties of atmospheric aerosols at the global scale, acquired by monitoring ground-
based networks and satellite missions, are needed in order to reduce these uncertainties and, consequently, improve our
knowledge of the effects of atmospheric aerosols on climate change.

Aerosols are unevenly distributed, both horizontally and vertically, with significant concentrations over landforms such as
deserts (e.g. Laurent et al., 2010; Heinold et al., 2011; Ansmann et al., 2011) or large populated urban areas (e.g. Landulfo et
al., 2003; Sun et al., 2004). Aerosol particles frequently travel through the troposphere (e.g. Guerrero-Rascado et al., 2009;
Preißler et al., 2011; Sicard et al., 2012a, 2012b; Pereira et al., 2014; Granado-Muñoz et al., 2016) and exceptionally at the
lower stratosphere (e.g. Ansmann et al., 1997, 2018; Sawamura et al., 2012; Baars et al., 2019). Atmospheric aerosols and
clouds cause a strong radiative forcing, playing an important role in the climate system of their source region, as well as in
the regions over which they are transported (Stocker et al., 2013). Thus, a good understanding of atmospheric dynamics is
required. Nowadays, Global Observing Systems (GOS) allow the study of a great variety of atmospheric properties through
ground-based instruments and satellites. Ground-based instruments for aerosol monitoring are set at local stations, which are
unequally distributed in space, grouped in federated networks such as AERONET (Holben et al., 1998), EARLINET
(Pappalardo et al., 2014), LALINET (Guerrero-Rascado et al., 2016; Antuña-Marrero et al, 2017) and MPLNET (Welton et
al., 2006), and also unequally distributed in time, i.e. during intensive field campaigns, such as ACE (Bernath et al., 2005),
SAMUM (Heintzenberg, 2009), SAMUM-2 (Knippertz et al., 2011) and CHARMEX (Mallet et al., 2016), among others.
Satellites can observe large areas of the atmosphere and a few can provide a non-stop full coverage of the planet's
atmosphere and surface. Satellite missions enable the remote retrieval of a vast set of atmospheric and surface properties
such as multispectral images of surface reflectance (e.g. Claverie et al., 2018; Bioresita et al., 2018), atmospheric
composition (e.g. Veefkind et al., 2012), and detailed optical information of the atmosphere (e.g. Amiridis et al., 2015).



Nonetheless, until 2018, there was not a single satellite mission aimed to retrieve worldwide, continuous wind measurements from space.

The lack of worldwide real-time wind measurements turns out to be one of the main deficiencies of current GOS, affecting the reliability of numerical weather prediction (NWP) models and the analysis of the climate variability (WMO, 2004).
Aiming to sort out these limitations, and encouraging large-scale dynamic studies especially in remote regions, the European Space Agency (ESA) approved in 1999 the development of the Atmospheric Dynamic Mission Aeolus (ADM-Aeolus, now called Aeolus) mission, named after the Keeper of the Winds in Greek Mythology. The satellite was finally launched on the 22nd August 2018, with the Atmospheric Laser Doppler Instrument (ALADIN) on board. At that time, it became the fifth satellite in space of the ESA's Living Planet programme, the first European satellite with a lidar onboard and the first space-
borne Doppler lidar ever able to measure vertical wind profiles on a global basis.

The complete coverage of the planet and the variety of properties provided by Aeolus will imply an important improvement of numerical models (Stoffelen et al. 2006; ESA, 2008; Horányi et al., 2015a, 2015b), as well as a contribution to the understanding of atmospheric dynamics in the troposphere and lower stratosphere (Straume et al., 2020). During the 2020 pandemic of COVID-19, continuous near-real-time worldwide measurements from the Aeolus mission served as a prominent
remedy to the lack of wind measurements, especially in the high troposphere and lower stratosphere, caused by air traffic reduction (Ingleby, 2020).

Prior to the satellite launch, ESA promoted several campaigns for the intercomparison of the satellite's products, for both wind and optical retrievals (Straume et al., 2019) in order to determine possible observation biases and increase the data quality. ESA encourages the participation of organizations around the world, such as EARLINET, LALINET, NOAA
(National Oceanic and Atmospheric Administration of the United States of America) and several other nation-wide organizations in Europe, Asia and North America. The current study is engaged within EARLINET cal/val (calibration and validation) activities of the Aeolus mission campaign at the continental scale ('Aeolus L2A aerosol and cloud product validation using the European Aerosol Research Lidar Network EARLINET', ref. 5166), promoted by the ESA (Straume et al., 2019). The cal/val activities encourage independent stations in the framework of EARLINET to perform ground-based
simultaneous measurements with Aeolus overpasses for validation activities. The present paper is one of the few devoted to the intercomparison of optical products from Aeolus at a continental scale. The geographic and temporal coverage of EARLINET stations along the quality-assured measurements provides an excellent framework for the intercomparison of Aeolus products under different atmospheric conditions and aerosol concentrations. Thus, the continuation of the cal/val activities are of great importance.
Several cal/val campaigns took place in the framework of EARLINET in the last decades, such as the one (still ongoing) for the Cloud-Aerosol Lidar and Infrared Pathfinder Satellite Observations (CALIPSO) (Winker et al., 2007), the first satellite mission focused on monitoring vertically-resolved aerosol and cloud optical products worldwide funded by NASA (e.g. Mamouri et al. 2009; Pappalardo et al., 2010; Amiridis et al., 2010, 2015; Papagiannopoulos et al., 2016). Also, the evaluation of aerosol optical products from the Cloud-Aerosol Transport System (CATS) onboard the International Space





Station (ISS) (Yorks et al., 2015, 2016; Rodier et al., 2016) was performed within the EARLINET community (Proestakis et al., 2019). Regarding Aeolus, several cal/val studies have focused on Aeolus wind products (known as L2B products), including wind profiles and their biases, with frequent intensive campaigns (e.g. Baars et al., 2020; Lux et al., 2020; Witschas et al., 2020, Guo et al., 2020). Recently, some applications of Aeolus optical products (known as L2A) have been reported (Dai et al., 2021; Feofilov et al., 2021). However at the time of writing this article, no studies assessing the

calibration and validation of Aeolus aerosol products with ground-based stations have been published.

The present study presents the intercomparison of Aeolus L2A aerosol optical products, in particular, the particle backscatter coefficients at 355 nm, with analogous ground-based lidar measurements from three different ACTRIS/EARLINET stations in the Iberian Peninsula, namely Granada, Évora and Barcelona. This article is structured as follows. Section 2 presents the experimental setup of the different systems, addressing the characteristics of Aeolus and ALADIN, along with the main

aspects of the ground-based ACTRIS/EARLINET stations involved in the study. Section 3 is devoted to the methodology followed in the intercomparison process, the peculiarities of Aeolus products and ground-based measurements, as well as the criteria set for the comparison. Section 4 gathers the results and discussion. Finally, section 5 summarizes the main findings of the study, with special attention to the most relevant aspects for the satellite mission.

## 2 Experimental setup

### 2.1 Aeolus satellite

Aeolus was launched to space on the 22nd August 2018 from the French Guiana. It is placed on a Sun-synchronous orbit at an altitude of 320 km and inclination angle of 97°. It moves at 7.71 km s$^{-1}$ and completes an orbit in 90 minutes (16 orbits per day) (Reitebuch et al., 2018; Flamant et al., 2020). The revisit time is 7 days (Flamant et al., 2020). The orbit setting of Aeolus allows it to overpass the equator at 06:00 and 18:00 Local Solar Time (LST) during ascending and descending

modes, respectively, in order to minimize the background noise caused by solar radiation, as it avoids solar zenith.

The Aeolus satellite is equipped with a single instrument, i.e. the ALADIN lidar. The instrument, based on the Doppler wind lidar technique, acquires profiles of wind speed and particle optical properties in the troposphere and low stratosphere up to 30 km (Ansmann et al., 2007; Flamant et al., 2008). The ALADIN lidar points towards the Earth's surface with an angle of 35° compared to nadir, although due to the planet's curvature this angle changes to 37.6° at surface level (Reitebuch et al.,

2018). Due to the orbit configuration and the instrument design, ALADIN only retrieves the projection of the horizontal wind speed over the line-of-sight (HLOS), a variable sufficient for the characterization of the wind field (ESA, 2008). This variable is known as Level 2B (L2B) product. As spin-off products, ALADIN also provides particle optical properties, namely the particle backscatter and extinction coefficients among others, known as Level 2A (L2A) products, which are retrieved through the Standard Correct Algorithm (SCA), Standard Correct Algorithm middle bin (SCAmb), Iterative

Correct Algorithm (ICA) and Mie Channel Algorithm (MCA) separately (Flamant et al., 2020). The retrieval employs the High Spectral Resolution Lidar (HSRL) technique (Wandinger, 1998). Additionally, L2C products consist of wind fields



after the assimilation of L2B profiles by the forecast models of the European Center for Medium-Range Weather Forecasts (ECMWF) (Ingmann an Straume, 2016). On 12th May 2020, L2B products were made available for the general public (aeolus-ds.eo.esa.int) after going through bias correction procedures. Currently, L2A products access is still limited until a
more confident version of the data products is achieved.

The instrument emits UV radiation at 355 nm and acquires the backscattered radiation through a dual receiver, consisting of two spectrometers, with Rayleigh and Mie channels, for molecule and particle backscattering, respectively (Ingmann and Straume, 2016). Consequently, two independent wind profiles can be retrieved. However, a single measurement of aerosol optical products is obtained from the combination of both signals. The emitted radiation is circularly polarized whilst the
receiver only detects the parallel (co-polar) component, resulting in underestimated particle backscatter coefficients and overestimated extinction coefficients (Flamant et al., 2020), especially under conditions with highly depolarizing particles. ALADIN emits light pulses at a repetition frequency of 50.5 Hz. A single observation, i.e. profile, is generated by averaging shots over a 12 s period corresponding to an horizontal resolution of 87 km. Each profile is divided into 24 vertical bins. The vertical resolution of each bin depends on the altitude: 500 m between 0 and 2 km (roughly the atmospheric boundary layer),
1 km between 2 and 16 km (roughly free troposphere) and 2 km between 16 and 30 km (roughly the lowermost stratosphere) (Ingmann and Straume, 2016).

**2.2 ACTRIS/EARLINET stations**

The EARLINET (European Aerosol Research Lidar Network; Pappalardo et al., 2014) network, in the framework of ACTRIS (Aerosols, Clouds, and Trace gases Research Infrastructure Network; actris.eu), aims to generate a vast database of
quality-assured vertical profiles of aerosol measurements under homogeneous standards around Europe. Thanks to its large spatial coverage of the European continent, ACTRIS/EARLINET actively has participated and participates in the validation/calibration of satellite missions measurements (Pappalardo et al., 2010; Amiridis et al., 2015; Papagiannopoulos et al., 2016; Proestakis et al., 2019). In this study, three ACTRIS/EARLINET lidar stations from the Iberian Peninsula, namely Granada, Évora and Barcelona, are considered.

The ACTRIS/EARLINET Granada station (37.164ºN, 3.605ºW, 680 m asl) is located in the Southeastern part of Spain, in a fairly populated region. The city lies in a geographic depression, at the foot of Sierra Nevada, with altitudes up to 3479 m asl to the east of the station. Aerosol particles from anthropogenic origin can be detected during the whole year, mainly released by fossil fuel burning (Lyamani et al., 2006, 2010, 2012). Due to the station proximity to the north of Africa, mineral dust intrusions from the Sahara Desert are often detected during the year, mainly in the summer season (Guerrero-Rascado et al.,
2008, 2009; Bravo-Aranda et al., 2015; Granado-Muñoz et al., 2016; Mandija et al., 2016), although winter dust intrusions are more frequent in the last years (Cazorla et al., 2017; Fernández et al., 2019). The concentration of continental aerosols from the European continent are also significant along the year (Lyamani et al., 2010). Notable events of local wildfires smoke (Alados-Arboledas et al., 2011) and long-range transported smoke from North-America (Ortiz-Amezcua et al., 2014, 2017; Sicard et al., 2019) are usual. Bioaerosol particle concentrations, especially pollen grains, are significant over the city





at specific periods of the year (Cariñanos et al., 2021). The station is equipped with a multispectral Raman lidar system, MULHACEN (LR331D400, Raymetrics S.A.), operated by the Atmospheric Physics research group in the Andalusian Institute for Earth System Research (IISTA-CEAMA) of the University of Granada. The lidar system is based on a Nd:YAG radiation source with a receiver at 355, 532 and 1064, as elastic channels, as well as 354 ($N_2$), 407 ($H_2O$) and 530 ($N_2$) nm as rotational and/or vibrational Raman channels, with a repetition frequency of 10 Hz. A polarization cube in the 532-nm optical

path enables to split the parallel (532p) and perpendicular (532s) components. Furthermore, the lidar has a nominal vertical and temporal resolution of 7.5 m and 1 min, respectively, with a full overlap height at around 800 m agl for all channels (Guerrero-Rascado et al. 2010; Navas-Guzman et al., 2011). The depolarization channel is routinary calibrated using the $\pm45°$ method (Bravo-Aranda et al., 2013). Further details about MULHACEN and subsequent upgrades can be found in Guerrero-Rascado et al. (2008, 2009) and Ortiz-Amezcua et al. (2020).

The ACTRIS/EARLINET Évora station (38.568°N, 7.912°W, 293 m asl) is located in the Southern part of Portugal, at around 100 km from Lisbon and the border with Spain, in a mainly flat and rural region with relatively low industrialization and low concentrations of anthropogenic aerosol (Pereira et al., 2009; Preißler et al., 2013). Smoke particles from nearby wildfires or industrialized regions are regularly transported over the station (Preißler et al., 2013; Pereira et al., 2014), as well as long-range transport from North America (Sicard et al., 2019; Baars et al., 2019). Furthermore, due to the proximity to the

Sahara Desert, mineral dust layers are frequent over the city (Pereira et al., 2009; Preißler et al. 2011, 2013) with many extreme events (Preißler et al., 2011; Valenzuela et al., 2017; Couto et al., 2021). Rare events of simultaneous biomass burnings and mineral dust intrusions have been characterized (Salgueiro et al., 2021). Significant bioaerosol concentration events can occur at specific stages of the year (Galveias et al., 2021). The lidar station is operated by the Institute of Earth Sciences, associated with the University of Évora. Among other atmospheric research instruments, the station holds a

multispectral Raman lidar of the POLLY$^{XT}$ type (Baars et al., 2016) named PAOLI. The Raman lidar is based on a Nd:YAG laser source, which enables it to emit radiation on 355, 532 and 1064 nm with a repetition frequency of 20 Hz. The same wavelengths are detected in reception along with two Raman channels at 387 ($N_2$) and 607 ($N_2$) nm. A depolarization filter allows the instrument to obtain the perpendicularly polarized component of the backscattered signal at 532 nm. Nominal measurements are acquired with a vertical and temporal resolution of 30 m and 30 s, respectively. The full overlap height is

around 800 m agl. Further details can be found in Preißler et al. (2011).

The ACTRIS/EARLINET Barcelona station (41.393°N, 2.120°E, 115 m asl) is located on the northeastern coast of Spain, in a highly populated and industrialized region. Due to its location, the different types of present aerosols are significantly diverse. The background aerosol load, mostly made of urban, traffic-related particles and marine aerosols, is located in the lowermost part of the boundary layer (Sicard et al., 2011). Saharan mineral dust intrusions are frequent along the year due to

its relative proximity to North Africa (Pérez et al., 2006). The variability of the aerosol optical properties in Barcelona in the atmospheric column has been studied by Sicard et al. (2011). The Barcelona lidar system is developed and operated by the Remote Sensing Laboratory of the Department of Signal Theory and Communications at the Universitat Politècnica de Catalunya. The system employs a Nd:YAG laser emitting pulses at 355, 532 and 1064 nm at a repetition frequency of 20 Hz





and linear polarization. In reception, the current configuration includes three elastic channels at 355, 532 and 1064 nm, one pure-rotational Raman channel at 354 nm, one vibro-rotational channel at 607 nm and one water vapor channel at 407 nm. In addition the system has also two channels for depolarization measurements at 355 and 532 nm, measuring the light passing through linear polarizers perpendicularly aligned with respect to the linear polarization sent by the laser. The measurements are acquired with a vertical resolution of 3.75 m and the approximate full overlap height being between 400 and 500 m agl. General details about the system can be found in Kumar et al. (2011); details about the depolarization channel set up and the pure rotational retrievals can be found in Rodríguez-Gómez et al. (2017) and Zenteno-Hernández et al. (2021), respectively.

The variety of locations, surroundings and orography of the three stations enables the study to explore coastal/inland, rural/urban and flat/mountainous effects on the quality of the comparisons between the ground station and the satellite. Aeolus overpasses Évora during an ascending mode (south to north) at around 52 km east every Friday at 18:17 UTC. In the case of Barcelona, the satellite overpasses the station during an ascending mode at 26 km west every Tuesday at 17:39 UTC. The station in Granada lies at the intersection of two Aeolus overpasses: every Thursday at 06:24 UTC (ascending orbit) and 18:04 UTC (descending orbit), both of them at about 24 km west. All three stations fulfill ESA's requirement that only satellite overpasses with ground-track distance less than 100 km should be considered (Straume et al., 2019). Figure 1a shows Aeolus overpasses over a wide part of Europe. Figure 1b is a zoom over the Iberian Peninsula where all three stations are located, as well as the associated Aeolus overpasses of the case studies analyzed in section 4.2.

## 3 Methodology

### 3.1 Database and intercomparison methodology

For the intercomparison between ground-based measurements and satellite products, a series of spatio-temporal criteria was established taking into account the location of each station and the overpasses times. On the one hand, the Aeolus observation closest in distance to the station was chosen. For the location of the Aeolus overpass, the middle point of the 87 km horizontal average forming each single profile was considered. On the other hand, the temporal co-location was established according to the measurement protocols at each station. For Granada, a 1.5-hour interval containing the morning overpass time (i.e. 05:30 - 07:00 UTC) and a 1-hour interval containing the evening overpass time (i.e.17:30 - 18:30 UTC) were chosen. For Barcelona, a 1-hour range centered at the overpass time (i.e. 17:09 - 18:09 UTC) was considered. For Évora, a 1.5-hour interval containing the overpass time (i.e. 17:30 - 19:00 UTC) was considered to take into account the larger distance between the Aeolus ground track and the lidar site.

Aeolus measurements are generated under specific data processing, called baselines, which are constantly being improved and updated. In October 2020 Aeolus products from July 2019 to December 2019 and from 20th April 2020 to 6th October 2020 were reprocessed under a single baseline, Baseline 10 (B10), aiming to homogenize the processing of the products and



encourage Aeolus cal/val teams to proceed with B10 products. The Aeolus database considered in this work is exclusively composed of B10 Aeolus products and covers different seasons and atmospheric conditions.

The current study is developed under cloud-free scenarios. Cloud screening was performed by visual inspection of the ground-based profiles. Aeolus observations have been carefully and individually checked, and atmospheric conditions have been studied along each overpass to ensure cloud-free conditions.

Aeolus products are automatically processed by ESA. In the current study, only aerosol products (L2A) are considered, and in particular particle backscatter coefficients derived from the Standard Correct Algorithm (SCA) and Standard Correct Algorithm middle bin (SCAmb). These algorithms employ the information from the Rayleigh and Mie channels and the derived profiles are divided into a series of vertical bins. The difference between them is that SCAmb bins (or middle bins) are obtained from two halves of adjacent original SCA bins, aiming to reduce noise in the products. These products come

with quality flags that mark individually the validity of each bin measurement. The quality flags assess the signal-to-noise ratio of both Mie and Rayleigh channels, as well as retrieval uncertainties (known as error estimates). A full description of L2A products and their implemented algorithms is given by Flamant et al. (2020).

The ground-based measurements are processed by the Single Calculus Chain (SCC) (D'Amico et al., 2015, 2016; Mattis et al., 2016), the standardized tool that allows to automatically process the lidar data acquired at each station within

EARLINET. Very few measurements of the Barcelona station were not inverted successfully by the SCC. In those cases, and after checking the cloud-free condition, the measurements were inverted manually with an algorithm validated in previous algorithm intercomparisons at network level (Böckmann et al. 2004; Pappalardo et al., 2004; Sicard et al., 2009). All data used in this work are level 1.0 and 2.0 data from the EARLINET/ACTRIS database (actris.nilu.no).

### 3.2 Aeolus-like conversion of ground-based lidar particle backscatter coefficients

Aeolus underestimates particle backscatter coefficient as the receiver only detects the co-polar component of circular polarized backscattered radiation at 355 nm (Flamant et al., 2020), while ground-based lidars retrieve particle backscatter coefficients with the total backscattered radiation at 355 nm or 532 nm. For comparison purposes, the co-polar component of the ground-based observations at 355 nm must be extracted from the total particle backscatter coefficient, through the expression:

$$\beta_{co,355}^{part} = \frac{\beta_{total,355}^{part}}{1 + \delta_{circ,355}^{part}} \tag{1}$$

where $\beta_{co,355}^{part}$ is the co-polar component of the particle backscatter coefficient at 355 nm (henceforth labeled as Aeolus-like coefficient, $\beta_{Aeolus\ like,355}^{part}$), $\beta_{total,355}^{part}$ is the total component of the particle backscatter coefficient at 355 nm derived from the ground-based lidar, and $\delta_{circ,355}^{part}$ is the circular particle depolarization ratio at 355 nm, which is not directly measured by the considered ground-based lidars. The linear particle depolarization ratio can be easily converted into circular particle

depolarization ratio from (Mishchenko and Hovenier, 1995):


$$\delta_{circ,355}^{part} = \frac{2\delta_{linear,355}^{part}}{1 - \delta_{linear,355}^{part}} \qquad (2)$$

where $\delta_{linear,355}^{part}$ is the linear particle depolarization ratio at 355 nm. The stations in Évora and Granada do not hold a depolarization channel at 355 nm and lack the possibility to measure $\delta_{linear,355}^{part}$. A conversion of $\delta_{linear,532}^{part}$ in $\delta_{linear,355}^{part}$ is proposed in the form of:

$$\delta_{linear,355}^{part} = K_\delta \cdot \delta_{linear,532}^{part} \qquad (3)$$

where $K_\delta$ is the spectral conversion factor. Thus, a thorough bibliographic review of previous multispectral depolarization studies has been conducted and discussed in Section 4.1 to estimate such a conversion factor. The third station, Barcelona, does measure both depolarization ratios but for the sake of consistency of the data processing, Barcelona $\delta_{linear,355}^{part}$ was calculated the same way than the other two stations. In Section 4.1 the measurements of $\delta_{linear,355}^{part}$ and $\delta_{linear,532}^{part}$ in

Barcelona are superimposed onto the literature results in order to validate our methodology.

### 3.3 Statistical parameters

A key point of the intercomparison is the vertical resolution of each profile. Aeolus divides each profile in a set of 24 vertical bins not homogeneously distributed. The resolution of these bins depends on the altitude range: 500 m between 0 and 2 km asl, 1 km between 2 and 16 km asl and 2 km between 16 and 30 km asl (Ingmann and Straume, 2016). Because the ground-

based lidars present a much finer resolution, of the order of a few meters, the resolution of each ground-based profile has been degraded to the Aeolus vertical resolution. Thus, the different ground-based vertical values within a given Aeolus bin are averaged into a single value, permitting a bin-to-bin intercomparison. This degradation process is performed on the ground-based Aeolus-like profiles in the last stage, prior to the statistical analysis calculations. Ground-based vertical coverage depends on the station and on each particular case, typically up to 10 km or up to the top of the highest aerosol

layer, while Aeolus profiles extend way beyond 10 km. Hence, only statistical comparisons below 10 km, where most of the aerosols are, is presented.

The statistical results are presented in vertical ranges of 1 km. A pair of values Aeolus/Aeolus-like will fall in a given 1 km range if the middle point of the bin lies within the vertical range (for instance, if a bin ranges from 1900 m to 2400 m, its middle altitude 2150 m lies within the 2 km vertical range). Three statistical parameters are calculated to assess the

intercomparison results: bias, root-mean-square error (RMSE) and linear fit. Bias profiles are calculated as follows:

$$bias\ (r) = \sum_{}^{N}\left(\beta_{Aeolus,355}^{part}(z) - \beta_{Aeolus\ like,355}^{part}(z)\right)$$

where $r$ is the vertical range, $z$ is the middle altitude of the bin's range that lies within the $r$ vertical range and $N$ is the number of pairs of values Aeolus/Aeolus-like whose $z$ lies within $r$. This parameter indicates if Aeolus underestimates ($bias\ (r) < 0$) or overestimates ($bias\ (r) > 0$) the co-polar particle backscatter coefficient in each region $r$. The RMSE

profile is also obtained as follow:





$$RMSE\ (r)\ =\ \sqrt{\frac{\sum^{N}\left[\beta_{Aeolus,355}^{part}(z)\ -\ \beta_{Aeolus\ like,355}^{part}(z)\right]^{2}}{N}}$$

Finally, the linear regression of $\beta_{Aeolus\ like,355}^{part}(z)$ against $\beta_{Aeolus,355}^{part}(z)$ is performed under a series of different settings, in order to test if they lie close to the 1:1 relation. The Pearson correlation coefficient, $R$, is calculated in all cases.

## 4 Results and discussion

### 4.1 Estimation of the depolarization spectral conversion factor from ground-based profiles to Aeolus-like products

For the calculation of the ground-based Aeolus-like profile (Eq. 1), $\delta_{linear,355}^{part}$ is needed (Eq. 2). PAOLI (Évora) and MULHACEN (Granada) hold only one depolarization channel at 532 nm, while the lidar system at Barcelona retrieves depolarization information at 355 and 532 nm channels. Therefore, the estimation of $\delta_{linear,355}^{part}$ from $\delta_{linear,532}^{part}$ is required. Pairs of ($\delta_{linear,355}^{part}$, $\delta_{linear,532}^{part}$) obtained from a thorough review of the literature and for different aerosol types are listed in Table 1. The literature provides a modest, but significant, dataset for different well characterized aerosol types, including mineral dust (fresh, aged, mixed), marine and mixed anthropogenic. These three aerosol types are the predominant aerosol types in Barcelona and the pairs ($\delta_{linear,355}^{part}$, $\delta_{linear,532}^{part}$) from the literature will be compared to measurements in Barcelona. Although not listed in Table 1, the literature also offers data for volcanic, bioaerosol and smoke particles.

A linear fit has been applied to the pairs ($\delta_{linear,355}^{part}$, $\delta_{linear,532}^{part}$) in order to set up a simple relationship to estimate $\delta_{linear,355}^{part}$ from $\delta_{linear,532}^{part}$ (Eq. 3) through the factor $K_{\delta}$, called the depolarization spectral conversion factor. Figure 2a shows the scatterplot of ($\delta_{linear,355}^{part}$, $\delta_{linear,532}^{part}$) for dust, marine and mixed anthropogenic aerosol from the literature, that are the aerosol types present in the cases used for the intercomparison. The best linear fit for these types together is obtained for $K_{\delta} = 0.82 \pm 0.02$, with a fairly acceptable statistical significance (Pearson correlation coefficient $R = 0.99$). The scatterplot for the other aerosol types (volcanic, bioaerosol and smoke) is shown in Figure 2b, and the linear fit is calculated separately for each aerosol type. For biomass burning particles, $\delta_{linear,355}^{part}$ is higher than $\delta_{linear,532}^{part}$ and $K_{\delta} = 1.36 \pm 0.08$, with a reliable statistical significance ($R = 0.97$). For bioaerosols, $K_{\delta}$ is significantly smaller ($0.46\pm0.04$), and less significant ($R = 0.91$). For volcanic particles, $K_{\delta} = 0.82 \pm 0.13$ (and $R = 0.98$). The large variability in biomass burning depolarization is related to the aging of smoke particles, while for bioaerosol particles it comes from the wide variety of bioaerosols (see e.g. Cao et al. (2010) from where most of the bioaerosol data come from in Figure 2b). Multispectral studies for volcanic particles are currently scarce in the literature.

Figure 2c is the same as Figure 2a with the addition of dust and non-dust cases measured with the dual-polarization system in Barcelona. The non-dust cases in Barcelona correspond generally to a mixture of marine and anthropogenic particles. As it can be seen, the measurements are consistent with the literature. The fitting of the full dataset (literature values plus



Barcelona measurements) provides a spectral conversion factor $K_\delta = 0.76 \pm 0.01$ (and $R = 0.99$). Thus, the literature-
derived relation for dust, marine and mixed anthropogenic is corroborated by the experimental values acquired over the
Iberian Peninsula.

The relationships obtained between $\delta^{part}_{linear,355}$ and $\delta^{part}_{linear,532}$ (dust plus non-dust, biomass burning, volcanic and
bioaerosols; see Figure 2) aim to serve as a look-up table for any station where only the depolarization channel at 532 nm is
available, which is a frequent handicap for many lidar systems worldwide. In the case of the Aeolus overpasses considered in
our study, only dust and non-dust (marine and anthropogenic) particles have been identified, with no evidence of smoke,
volcanic or bioaerosol particles in significant concentrations. For dust and non-dust types, $K_\delta$ equals to $0.82 \pm 0.02$ and is
implemented from now on in the calculation of the Aeolus-like profile.

### 4.2 Case studies

A set of case studies are given for the different stations under relevant atmospheric conditions. These case studies illustrate
the intercomparison process and serve as graphic examples of the Aeolus performance. Sun-photometer measurements are
taken into account for the sake of completeness aerosol typing, through the study of the aerosol optical depth at 675 nm
($AOD_{675}$), the AOD-related Ångström exponent calculated between the channels at 440 and 870 nm ($AOD\text{-}AE_{440\text{-}870}$), the
fine/coarse mode AOD fraction at 500 nm, the particle size distribution and the single scattering albedo at 440 and 1020 nm
($SSA_{440}$ and $SSA_{1020}$) (e.g. Dubovik et al., 2002; Gobbi et al., 2007; Lee et al., 2010; Shin et al., 2019; Foyo-Moreno et al.,
2019). AERONET level 1.5 or level 2.0 products, depending on availability, computed from the version 3 algorithm (Giles
et al., 2018) are used.

The location of the stations is highly interesting due to their proximity to the Sahara Desert and mainland Europe, so
frequent events of mineral dust and anthropogenic particles could be detected by the satellite. In addition, Barcelona lies just
in the coastline, and both Barcelona and Granada present high concentrations of anthropogenic aerosol, while Évora aerosol
concentrations could be classified as rural. Thus, Aeolus operation can be tested under a complete set of atmospheric
scenarios.

### 4.2.1 Case study of anthropogenic aerosol: Granada, 5th September 2019

Aeolus overpassed Granada at 18:04 UTC on the 5th September 2019 with a horizontal distance of 14 km (from Aeolus
observation middle point). The daily range corrected signal time series and the ground track of the satellite are presented in
Figures 3a and 3b, respectively. As observed in the time series, a significant particle concentration is detected during the
whole day up to 4 km asl approximately, with a few cirrus clouds between 6 and 14 km asl. Thus, following the ESA
requirements, and for the sake of homogeneity, this case is not included in the statistical analysis presented in Section 4.3,
but due to its interesting features it is included as a case study of Aeolus performance. Figure 3b shows Aeolus SCAmb





backscatter along the orbit, considering Aeolus quality flags. A significant aerosol layer over North Africa and the South of
the Iberian Peninsula can be observed.

The HYSPLIT model (Figure 4) reveals that the air masses over Granada at 18:00 UTC come from mainly two differentiated
regions: above roughly 4.3 km agl (equivalent to 5 km asl, not shown) the air masses traveled the Atlantic from North
America, while below 4.3 km agl the air masses were mostly stagnant over the Iberian Peninsula. Thus, mostly
continental/anthropogenic particles are expected over Granada in the lowermost region.

The measurements from the co-located Sun-photometer (not presented here) suggest a predominance of fine mode particles
during the whole available period. The AOD-AE$_{440-870}$ values agree with the presence of small particles
(continental/anthropogenic aerosol), with a mean value of $1.30 \pm 0.08$. Furthermore, the AOD$_{675}$ slightly varies throughout
the day, with a mean value of $0.15 \pm 0.01$. The columnar particle size distribution displays a bimodal distribution in the early
morning with a mean effective radius of $0.39 \pm 0.06$ μm and the SSA around 0.99 for all wavelengths, indicating the
presence of non-absorbing particles (Shin et al., 2019).

Figure 5 presents the most relevant vertically-resolved quantities measured by Aeolus and the ground-based lidar system in
Granada. According to the particle backscatter coefficients at 355 and 532 nm (Figure 5a) a significant aerosol layer is
observed up to 4 km approximately and a clear free troposphere above with a thin cirrus cloud at around 11 km asl. The
depolarization channel at 532 nm (Figure 5b) indicates the homogeneity of the depolarizing particles within the layer, with
low values of $\delta^{part}_{linear,532}$. Furthermore, the backscatter-related Ångström exponent profile calculated with the 355 and 532 nm
lidar channels (not presented here) exhibits values around 2 for the whole layer and yields the presence of
anthropogenic/continental aerosol and the absence of mineral dust particles. The large values of backscatter coefficients and
Ångström exponent together with low values of $\delta^{part}_{linear,532}$ corroborate the presence of non-depolarizing anthropogenic
particles in a significant aerosol layer. First, the Aeolus satellite properly detects the layer under both SCA and SCAmb
(Figure 5c), in terms of co-polar particle backscatter coefficient values and vertical layering, with an excellent agreement for
these bins. Second, for this particular case, SCAmb retrievals present a better agreement with ground-based measurements
than SCA retrievals. Third, the satellite performance presents a surface-related effect for their lowermost bins, retrieving a
large and unreasonable co-polar particle backscatter coefficient. Fourth, the cirrus cloud shown in Figure 3a is well retrieved
by SCA but not by SCAmb. The implementation of Aeolus quality flags (Figure 5d) produces a notable decrease in the
amount of available data points. In this case, quality flags do not seem to help in cloud screening of the satellite data (at 11
km asl, Aeolus retrievals are the same with and without quality flags). However, SCA retrievals are improved since the
quality flags exclude the negative particle backscatter coefficients found between 6 and 9 km asl.

### 4.2.2 Case study of mineral dust: Évora, 28th June 2019

Aeolus overpassed Évora at 18:17 UTC on the 28th June 2019 with a horizontal distance of 61 km (from Aeolus observation
middle point). The time series of the daily lidar range corrected signal at 355 nm and the ground track of the satellite are





presented in Figure 6a and 6b, respectively. A notable and homogeneous layer can be identified throughout the whole day below 3 km asl. Furthermore, there is no evidence of cloud presence above the station. Figure 6b shows the Aeolus SCAmb backscatter retrievals along the orbit with the quality flags applied. A homogeneous layer can be seen over the western side of the Iberian Peninsula, with a wider vertical extension over Morocco.

The HYSPLIT model indicates that the 12:00 UTC air masses over Évora at 1.7 and 2.7 km agl (equivalent to 2 and 3 km asl) are coming directly from lower altitudes in Northern Africa (Figure 7a). The back trajectories of the air masses over Évora at 18:00 UTC (Figure 7b), closer in time to Aeolus overpass, still indicate an origin over lower altitudes in the African continent for the air masses at 1.7 km agl but no longer for 2.7 km agl. Both BSC-DREAM8b and NAAPS models (not shown here) indicate the presence of low but non-negligible concentrations of mineral dust particles over the region at 18:00

UTC.

The co-located Sun-photometer measurements (not presented here) suggest the predominance of coarse-mode particles until 09:30 UTC approximately. From 09:30 UTC on, fine-mode particles dominate. The AOD-AE$_{440-870}$ agrees with the presence of a mineral dust layer over the station during the first half of the day, with a mean value of $0.70 \pm 0.07$. This value maintains between 0.88 and 1.12 approximately during the second half of the day, indicating that the dust episode is vanishing over

Évora. The AOD indicates that the possible mineral dust layer over the first half of the day does not present large concentrations of mineral particles, although these values are far from representing a clean atmosphere. The columnar size distribution endorses this hypothesis, with a large predominance of large particle radii during the morning and a decrease in the concentration after noon, with mean total effective radii of $0.60 \pm 0.02$ μm and $0.47 \pm 0.03$ μm, respectively. The SSA corroborates the presence of mineral dust during the day, with a positive difference SSA$_{1020}$-SSA$_{440}$ of +0.04 in the morning

and a difference of almost zero (-0.001) in the late afternoon. Thus, the models and the Sun-photometer measurements indicate the presence of a minor mineral dust episode over Évora that vanishes in the afternoon. The satellite overpass takes place at 18:17 UTC, when the dust episode is practically finished and the dust concentration is low. At that time, AOD$_{675}$ takes a value of 0.11 while AE reaches 1.04.

Figure 8 presents the most relevant vertically-resolved quantities measured by Aeolus and the ground-based lidar system in

Évora. A well defined layer in the lowermost atmosphere is detected by the lidar at both 355 and 532 nm channels (Figure 8a). The lidar $\delta^{part}_{linear,532}$ (Figure 8b) agrees with the presence of mineral dust particles mixed with other non-polarizing particles. Furthermore, the backscatter-related Ångström exponent profile calculated with the 355 and 532 nm channels (not presented here) takes values close to zero, corresponding to mineral dust particles (e.g. Müller et al., 2007; Guerrero-Rascado et al., 2009; Preißler et al., 2011; Fernández et al., 2019), in the whole vertical range of the detected layer. Aeolus detects this

layer under both SCA and SCAmb (Figure 8c). First, a fair agreement between the satellite and ground-based systems is achieved in the whole profile. Second, this case study leaves no doubts that the proposed Aeolus-like conversion of the total component ground-based backscatter profiles to co-polar component profiles has to be considered. Third, no significant difference is detected between SCA and the SCAmb intercomparison results for this case. Fourth, Aeolus behaves stably above the layer, measuring particle co-polar backscatter coefficients close to zero in the free troposphere, although sometimes




it retrieves negative and meaningless values. Fifth, the satellite presents a surface-related effect for the lowermost bin, retrieving large (and unrealistic) co-polar particle backscatter coefficients. In the final stage, quality flags are applied to Aeolus measurements (Figure 8d), presenting a notable decrease in the amount of available Aeolus values to perform the intercomparison. These quality flags limit the intercomparison to the layers with significant aerosol loads, preventing the intercomparison of Aeolus behavior in the free troposphere. Moreover, current preliminary quality flags do not prevent surface-related effects on the final Aeolus measurements.

### 4.2.3 Case study of smoke: Barcelona, 2nd July 2019

Aeolus overpassed the city of Barcelona at 17:39 UTC on the 2nd July 2019 with a horizontal distance of 35.17 km (from Aeolus observation middle point). Figure 9a presents the time series of the daily lidar range corrected signal at the 1064 nm channel, and the ground track of the satellite is presented in Figure 9b. A significant aerosol layer, which is itself stratified in thinner layers, is detected up to 2.5 km asl, as well as a sparse small layer above, between 2.5 and 4 km asl. Figure 9b also presents the Aeolus SCAmb co-polar backscatter retrievals along the orbit with the quality flags applied. Figure 9b displays Aeolus SCAmb co-polar backscatter coefficients along the considered orbit applying the quality flags. A significant layer is captured by the satellite above France, the north-western part of the Iberian Peninsula and the Mediterranean Sea.

The HYSPLIT model indicates that the air masses over Barcelona at 18:00 UTC between 1.9 and 2.9 km agl (equivalent to 2 km and 4 km asl), approximately, come directly from Southeastern France/northwestern Italy (Figure 10). In particular, the air masses at 1.9 km agl have the typical pattern of local recirculation and might carry pollutants from southern France. Additionally, the NAAPS model (not shown here) yields the presence of significant smoke concentrations over southeastern France/northwestern Italy during the previous days. Furthermore, both Aqua-MODIS and Terra-MODIS measurements (not shown here) reveal the existence of wildfires in southeastern France/northwestern Italy.

The co-located Sun-photometer retrievals indicate the predominance of fine mode particles throughout the day. The daily mean AOD-AE$_{440-870}$ is $1.43 \pm 0.13$, while the AOD$_{675}$ also remains constant throughout the day, with a mean value of $0.15 \pm 0.02$. The particle size distribution presents two distinct modes, with a mean effective radius of $0.39 \pm 0.04$ μm approximately, where the fine mode dominates. This distribution is constant over the day. After noon, all of the retrieved SSA$_{1020}$ lie below 0.95, suggesting the prevalence of absorbing particles (Shin et al., 2019). On the other hand, for all of the available sets of SSA$_{1020}$ and SSA$_{440}$, the difference SSA$_{1020}$-SSA$_{440}$ is negative, so the presence of mineral dust is discarded (Dubovik et al., 2002). These Sun-photometer measurements suggest the presence of a smoke layer over the station during the whole day. Aeolus overpassed the station at 17:39 UTC, when the values of AOD$_{675}$ and AOD-AE$_{440-870}$ are 0.15 and 1.47, respectively.

The most relevant vertically-resolved properties measured by Aeolus and the ground-based lidar in Barcelona are represented in Figure 11. It detects the presence of several layers: a non-depolarizing aerosol layer up to 2.5 km asl approximately and a depolarizing layer above 2.5 km asl (Figures 11a and 11b). From 2.5 to 6 km the particle backscatter coefficient decreases significantly while $\delta_{linear,532}^{part}$ increases with respect to the lower layer up to values close to 0.2. Such high values for smoke





particles have been observed recently by Khaykin et al. (2018), Haarig et al. (2018), Sicard et al. (2018) and Hu et al. (2019) but they were produced by aged smoke and observed at high altitudes, which is not the case here. Another possible and more
plausible explanation is the mixture of mineral dust and biomass burning, which also produces such high values of $\delta_{linear,532}^{part}$ (Groß at el., 2011a). Indeed, the Iberian Peninsula was hit unusually frequently by dust episodes during June-July 2019 (Córdoba-Jabonero et al., 2021). The backscatter-related Ångström exponent profile calculated with the 355 and 532 nm lidar channels (not presented here) exhibits values around 1.50 in the lower layer and 1.25 between 2.5 and 4 km asl. These values are in accordance with the values reported in the literature for Iberian smoke (Alados-Arboledas et al., 2011; Pereira et al.,
2014) and anthropogenic particles (Lyamani et al. 2006; Alados-Arboledas et al., 2011). First, the satellite presents a satisfactory agreement with the ground-based lidar in the whole available profile under both SCA and SCAmb (Figure 11c). Second, the satellite clearly detects a layer up to 5 km asl approximately. Third, the SCA retrieves stable close to zero values in the free troposphere, although some are negative and meaningless. Fourth, the surface-related effect is present in the lowermost bins. In this case, Aeolus quality flags (Figure 11d) seem to remove the surface-related effect, but again they do
not allow for investigating the Aeolus performance in the free troposphere.

### 4.3 Statistical analysis

This section assesses the intercomparison of Aeolus SCA and SCA middle bin products with ground-based measurements from a statistical point of view. The process is performed considering Aeolus quality flags to achieve a further understanding of the products. Moreover, ground-based measurements are cloud screened (Granada case study for the 5th September 2019,
see details in Section 4.2.1., is removed as well). Taking into account the requirements and considerations presented in Section 3, the initial database is largely reduced, in order to ensure the reliability of the intercomparison. From the initially available measurements, i.e. 101 B10-overpasses for Granada, 51 for Évora and 52 for Barcelona, and after applying the set of requirements, the intercomparison has been performed with 24 cases for Granada, 15 cases for Évora and 16 cases for Barcelona, leading to enough statistical significance.

First, we address the general performance of the satellite, with emphasis on the domain of Aeolus co-polar particle backscatter coefficient retrievals. On the one hand, Figure 12a shows that Aeolus SCA retrievals range from approximately -2 Mm$^{-1}$ sr$^{-1}$, to large and unrealistic values (up to 86 Mm$^{-1}$ sr$^{-1}$) which are associated with the surface-related effect shown by the satellite. On the other hand, Aeolus SCAmb retrievals range from 0 Mm$^{-1}$ sr$^{-1}$ to similarly large and unrealistic values (up to 68 Mm$^{-1}$ sr$^{-1}$) (Figure 12b). With the implementation of the quality flags (Figure 12c and 12d), all of the sets range from 0
Mm$^{-1}$ sr$^{-1}$. In fact, a little more than 1 out of every 3 SCA values (35.5 %) are negative. Additionally, unrealistic values are still flagged as valid after the application of the quality flags, although the amount is slightly diminished in absolute values but not in relative terms. In the case of the dataset without the quality flag filtering, the SCA presents 29 measures above 7 Mm$^{-1}$ sr$^{-1}$ (used here to delimit unrealistic values) (5.5 % of all 529 measures) while the SCAmb presents 25 (5.8 % of all 430 measures). After the filtering of the dataset, 28 values (27.5 % of all 102 measures) are reported for the SCA and 22
(27.5 % of all 80 measures) for the SCAmb. Regarding the unfiltered dataset, the number of values above 7 Mm$^{-1}$ sr$^{-1}$ is



negligible, although their effects can be observed in the statistical results of the lowermost regions, as will be addressed further on. On the contrary, the amount of values above 7 Mm⁻¹ sr⁻¹ is substantial and relevant for the filtered dataset as the number of values below 7 Mm⁻¹ sr⁻¹ is dramatically reduced.

Figures 13a and 13b present $\beta^{part}_{Aeolus\,like,355}$ and $\beta^{part}_{Aeolus\,SCA}$ values, which do not fit any linear or nonlinear relation. The
linear fitting of each dataset (red lines in Figure 13) is not good ($R$ smaller than 0.4 in all cases). Several types of relationship were tested and no valid model was found. Quality flagged data (Figures 13c and 13d) worsen the linear relationship ($R$ smaller than 0.25 in both cases). The same analysis has been performed for each station separately (not presented here) in order to search potential particularities of each site. Analogous and unsatisfactory results were found with the dataset of each station.

Aeolus backscatter coefficient uncertainties (known as Aeolus error estimates) are addressed through the biases between satellite and ground-based measurements. Figure 14 reveals that the larger the Aeolus uncertainties, the larger the bias. In this case, it can clearly be seen that quality flags implementation does not remove Aeolus retrievals with large uncertainties in absolute terms. Quality flags assess Aeolus errors relative to the backscatter coefficient retrievals. Therefore, the lowermost measures of Aeolus, which generally present large and unrealistic co-polar backscatter coefficients and large
uncertainties, are still flagged as valid. Furthermore, Figures 14a and 14b shows that the SCAmb retrievals present smaller errors.

### 4.3.1 Granada

The statistical results for Granada are obtained from the 24 selected cases (309 SCA data points and 246 SCAmb data points). Aeolus retrieves co-polar particle backscatter coefficients from approximately 32 km to the ground level (downward
view). However, due to the station's altitude (680 m asl) and the lidar full overlap height, no matching measurements are available between 0 and 1 km asl (Figure 15). On the one hand, Aeolus products present a significant surface-related effect for the lowermost regions, between 1 and 2 km asl. Thus, the satellite strongly overestimates the co-polar particle backscatter coefficient in the 1 to 2 km asl vertical range (with no quality flag implementation, Figures 15a, 15b, 15c and 15d), with a SCA bias around 9 Mm⁻¹ sr⁻¹ and RMSE around 19 Mm⁻¹ sr⁻¹ along with a SCAmb bias around 8 Mm⁻¹ sr⁻¹ and RMSE around
11 Mm⁻¹ sr⁻¹. This surface effect may affect as well the 2 to 3 km asl range to a lesser extent. Figures 15a and 15b show that the general performance of the SCA underestimates particle backscatter coefficients from 3 to 11 km asl, with a fair bias value, smaller than 0.4 Mm⁻¹ sr⁻¹ in any case, and RMSE lower than 1 Mm⁻¹ sr⁻¹ (average values of -0.18 ± 0.07 and 0.6 ± 02 Mm⁻¹ sr⁻¹ respectively). On the other hand, Figure 15c shows that the SCAmb does not present any trend between 2 and 11 km asl, with the bias values oscillating around 0, between -0.11 and 0.17 Mm⁻¹ sr⁻¹ (average value of 0.07 ± 0.11 Mm⁻¹ sr⁻¹).
For this algorithm, the RMSE (Figure 15d) lies below 0.7 Mm⁻¹ sr⁻¹ in every range above 3 km asl (average value of 0.40 ± 0.13 Mm⁻¹ sr⁻¹). SCAmb derived RMSE values are smaller than those obtained with the SCA in every vertical range. Thus, a better agreement is found between the satellite and the ground-based measurements with the SCAmb. Furthermore, with the





quality flags implementation (Figures 15e-h), the number of available measurements flagged as valid is largely diminished (only 1 out of 7 SCA values and 1 out of 6 SCAmb values), especially above 3 km asl. Therefore, the statistical significance
of the results is also reduced and the reliable results are limited to the lowermost ranges. Additionally, after the quality flags consideration between 2 and 4 km asl, i.e. the statistically significant ranges, SCA and SCAmb RMSE values increase a 45 and 61 %, respectively. The use of the quality flags worsens the average agreement between the satellite and the ground-based system and does not avoid the surface-related effect on the measurements.

### 4.3.2 Évora

In the case of Évora, the statistical results are derived from 15 selected cases (150 SCA data points and 108 SCAmb data points). Figure 16 indicates that a few matching measurements in the vertical range from 0 to 1 km asl could be found for this lidar system, due to the station's altitude (293 m asl). On the one hand, the surface-related effect is present in all cases in the lowermost regions as well, from 0 to 2 km asl, with the vertical range from 0 to 1 km clearly more affected by this effect. In these regions Aeolus largely overestimates co-polar particle backscatter coefficient (Figures 16a-d) with a SCA bias
around 11 Mm$^{-1}$ sr$^{-1}$ and RMSE around 13 Mm$^{-1}$ sr$^{-1}$ along with a SCAmb bias around 5 Mm$^{-1}$ sr$^{-1}$ and RMSE around 6 Mm$^{-1}$ sr$^{-1}$. Therefore, SCAmb retrieval is less affected by the surface effect. On the other hand, an inhomogeneous performance is observed for the SCA above 2 km asl (Figure 16a), with bias values ranging from -0.2 to 0.5 Mm$^{-1}$ sr$^{-1}$ (average value of 0.02 ± 0.24). Additionally, an average RMSE value of 0.5 ± 0.3 Mm$^{-1}$ sr$^{-1}$ is obtained for the vertical ranges above 2 km asl (Figure 16b). The SCAmb (Figures 16c and 16d) seems to overestimate particle backscatter coefficient from 2 to 11 km asl,
although a fair agreement with ground-based measurements is observed (average bias value of 0.11 ± 0.08 Mm$^{-1}$ sr$^{-1}$ and RMSE of 0.36 ± 0.17 Mm$^{-1}$ sr$^{-1}$). Therefore, the SCAmb retrievals present a better agreement with ground-based measurements than the SCA. The number of selected cases is smaller for Évora than the ones used in the case of Granada, but still statistically significant. However, with the implementation of the quality flags (Figures 16e-h) the amount of valid matching measurements is drastically reduced (only 1 out of 5 SCA values and 1 out of 6 SCAmb) and the results cannot be
considered statistically significant in any range. The only improvement in the agreement between the systems is observed between 1 and 2 km asl, but the statistical significance has to be taken into account. Again, the application of the quality flags does not avoid the surface-related effect on the final results.

### 4.3.3 Barcelona

The statistical results for Barcelona are derived from 16 selected matching cases (80 SCA data points and 76 SCAmb data
points). In Barcelona, the validation process of the particle backscatter coefficient cuts the profiles where the aerosol layers end, so the vertical coverage usually does not extend higher than 5 or 6 km asl (depending on the atmospheric conditions). Thus, no statistical intercomparison could be performed above this altitude. Furthermore, the station lies at a very low altitude above sea level (115 m asl) and its full overlap height (between 400 and 500 m agl) allows us to work with a significant amount of matching values between 0 and 1 km asl, the vertical range which is most affected by the surface



(Figure 17). Between 0 and 1 km asl Aeolus largely overestimates co-polar particle backscatter coefficients (with no quality flag implementation, Figures 17a and 17b), with an approximate SCA bias of 15 Mm$^{-1}$sr$^{-1}$ and RMSE of 21 Mm$^{-1}$sr$^{-1}$ along with a SCAmb bias around 9 Mm$^{-1}$sr$^{-1}$ and RMSE around 18 Mm$^{-1}$sr$^{-1}$. However, possible surface-related effects can be observed in the case of the SCA retrievals between 1 and 4 km asl (Figure 17a), where an average RMSE of $1.6 \pm 0.1$ Mm$^{-1}$ sr$^{-1}$ is observed. This could be explained by the complex terrain orography below the Aeolus ground-track, mostly affected

by the transition from sea to land with the Central Prelitoral System situated only 15 km from the coast and reaching almost 1000 m asl. On the contrary the SCAmb seems to be partially affected between 1 and 2 km asl (Figure 17b) and to a lesser extent (1.4 Mm$^{-1}$sr$^{-1}$). Therefore, the SCAmb is more robust to the surface effects. Nevertheless, Aeolus does not present a trend above 1 km asl neither under SCA, nor under SCAmb (Figures 17a and 17b), and the bias values ranges from -0.5 to 0.8 Mm$^{-1}$sr$^{-1}$ and from -0.2 to 0.3 Mm$^{-1}$sr$^{-1}$ respectively. In the rest of the available vertical ranges, Aeolus presents a slightly

better agreement with the ground-based system under SCAmb, with RMSE values below 0.5 Mm$^{-1}$sr$^{-1}$ between 2 and 7 km asl. Finally, when quality flags are applied (Figures 17c and 17d), the amount of valid matching measurements is reduced (almost 2 out of 5 SCA values and 2 out of 7 SCAmb values), affecting the statistical significance of the results. Additionally, after the quality flags consideration between 1 and 3 km asl, i.e. the statistically significant vertical ranges, SCA RMSE values increase a 40 % and SCAmb RMSE values a 65 % between 1 and 2 km asl. Thus, quality flag filtering of

the dataset worsens the statistical results and does not avoid the surface-related effect.

## 5 Conclusions

Aeolus satellite was launched in 2018. At the time of writing of this article, the longest, fully homogeneous product dataset has been reprocessed in baseline 10 (reprocessed products, B10 version). In this study we evaluated Aeolus B10 optical products with a thorough analysis of Aeolus co-polar backscatter coefficients under the standard correct algorithm (SCA)

and the standard correct algorithm middle bin (SCAmb), and an effective testing of Aeolus quality flags. This process was performed taking into account the ESA and the cal/val community recommendations through the intercomparison of Aeolus products with analogous ground-based measurements taken at the ACTRIS/EARLINET stations of Granada, Évora and Barcelona (Southwestern Europe), matching temporally and spatially the satellite's overpasses (55 cases). However, Aeolus overpasses at each station were analyzed separately, aiming to characterize Aeolus performance under different and relevant

atmospheric conditions, aerosol types and orographic features.

We assessed the so-called Aeolus-like conversion of ground-based measurements. Aeolus retrieves the co-polar component of the backscatter coefficient, which is not directly comparable to the total component measured at the surface stations. Thus, the co-polar component of the ground-based measurements has to be derived from the total one. In this work, an approach based on a thorough bibliographic review of dual-polarization measurements for relevant aerosol types, aiming to estimate

$\delta^{part}_{linear,355}$ from $\delta^{part}_{linear,532}$, was proposed. A relation of $\delta^{part}_{linear,355} = (0.82 \pm 0.02)\,\delta^{part}_{linear,532}$, which is endorsed by dual-polarization measurements in Barcelona, was found. Other cal/val teams are encouraged to take into consideration the



Aeolus-like conversion of the ground-based measurements, and the implementation of the $\delta^{part}_{linear}$ spectral relationship if needed, which has proven to be effective in our case studies.

Several types of linear and nonlinear relations were tested and no valid model was found for $\beta^{part}_{Aeolus\,like,355}$ and $\beta^{part}_{Aeolus}$.

Also, a relation between high Aeolus uncertainties and bias differences was noted. These results were observed at the three stations, suggesting that they were related to the satellite data characteristics and/or the methodology employed and difficulties inherent to satellite cal/val activities rather than to a specific feature of one particular station.

Aeolus SCAmb retrievals presented a better agreement with respect to ground-based measurements than the SCA ones. For the Granada station, the difference between algorithms was less significant, while the SCAmb presented much better results

than the SCA for the stations in Évora and Barcelona. Évora presented the highest agreement with the SCAmb retrievals, although the results for Barcelona and Granada were quite satisfactory as well. RMSE profiles obtained with the SCAmb are fairly similar for the three stations, providing consistency to the results obtained. Aeolus quality flags implementation entailed a strong reduction of the amount of Aeolus measurements initially available. For both SCA and SCAmb approximately 20 % of the data remains after the quality tests. This substantially affected the statistical significance of the

results of the filtered dataset. Additionally, Aeolus measurements over all of three stations presented a critical surface-related effect that caused Aeolus to drastically overestimate the co-polar backscatter coefficients. Depending on the station and the orography of the region, this effect extended up to higher altitudes. Finally, the statistical intercomparison was not improved after the quality flag application, e.g. the surface effect was not mitigated and even an increase of the RMSE (between a 40 and a 65 %) was observed.

It has been seen that under significant cloud conditions the satellite experiences saturation and retrieves inconsistent and invalid results. Even cirrus conditions can affect the results. However, the presented case study for Granada (5th September 2019) shows that the satellite is able to characterize thin cirrus clouds with a fairly acceptable agreement.

Despite the distance between the overpasses and the stations, and the fact that Aeolus products are generated by averaging horizontally over 87 km, a good agreement was found between Aeolus retrievals and ground-based lidar measurements,

demonstrating that Aeolus has a high potential for the worldwide characterization of the aerosol vertical distributions.

**Author contributions**

JAG, JLGR, MJC, MS: conceptualization, investigation, methodology, and validation; JAG, JLGR, MJC, JABA, MS, DBP, DB, MJGM, ARG, CMP, AC, POA, VS, MMJM, LAA: data curation and formal analysis; JAG, JLGR, MJC, MS: writing original draft; JAG, JLGR, MJC, JABA, MS, DBP, MJGM, ARG, AC, POA, LAA: review and editing.

**Competing interests**

The authors declare that they have no conflict of interest.



**Acknowledgements**

The analysis has been performed in the frame of the Aeolus Scientific Calibration & Validation Team (ACVT). The authors acknowledge the ESA project 'Aeolus L2A aerosol and cloud product validation using the European Aerosol Research Lidar
Network EARLINET and Cloudnet' (ref. Aeolus AO5166). ACTRIS-2 Research Infrastructure Project and Implementation Project of the European Union's Horizon 2020 research and innovation program (grant agreement No 654109 and 871115) as well as GRASP-ACE (GA 778349) are also acknowledged. This work is related to activities within the COST Action CA18235 PROBE (PROfiling the atmospheric Boundary layer at European scale). This work was also supported by the Spanish Ministry of Economy and Competitiveness (projects CGL2015-73250-JIN, CGL2016-81092-R, CGL2017-83538-
C3-1-R and CGL2017-90884-REDT), the Spanish Ministry of Science and Innovation (project PID2019-103886RB-I00), the Unity of Excellence "María de Maeztu" (project MDM-2016-0600) financed by the Spanish State Research Agency (AEI) and by the national Portuguese funds through FCT - Fundação para a Ciência e Tecnologia, I.P. (projects UIDB/04683/2020, UIDP/04683/2020, PTDC/CTAMET/29678/2017 and 0753_CILIFO_5_E). The authors thankfully acknowledge the FEDER program for the instrumentation used in this work and the University of Granada that supported
this study through the Excellence Units Program. Maria José Granados-Muñoz has received funding from the European Union's Horizon 2020 research and innovation programme under the Marie Skłodowska-Curie grant (agreement No 796539). Juan Antonio Bravo-Aranda received funding from the Marie Skłodowska-Curie Action Cofund 2016 EU project - Athenea3i grant (agreement no. 754446). Finally, the authors gratefully acknowledge the NOAA Air Resources Laboratory (ARL) for the provision of the HYSPLIT transport and dispersion model and/or READY website (ready.noaa.gov) used in
this publication.

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

| Aerosol type | $\delta^{part}_{linear,355}$ | | $\delta^{part}_{linear,532}$ | | Reference |
|---|---|---|---|---|---|
| | Mean | SD | Mean | SD | |
| Pure dust | 0.24 | 0.07 | 0.31 | 0.02 | Freudenthaler et al. (2008) |
| | 0.28 | 0.05 | 0.31 | 0.02 | |
| | 0.27 | 0.01 | 0.30 | 0.01 | Groß et al. (2011a) |
| | 0.25 | 0.03 | 0.30 | 0.01 | |
| | 0.20 | 0.04 | 0.240 | 0.045 | Groß et al. (2011b) |
| | 0.19 | 0.02 | 0.23 | 0.01 | |
| | 0.430 | 0.046 | 0.373 | 0.014 | Burton et al. (2015) |
| | 0.24 | 0.03 | 0.33 | 0.01 | Hoffer et al. (2020) |
| Aged dust | 0.30 | 0.05 | 0.34 | 0.02 | Wiegner et al. |





| | | | | |
|---|---|---|---|---|
| | 0.29 | 0.07 | 0.35 | 0.01 | (2011) |
| | 0.31 | 0.03 | 0.32 | 0.02 | |
| | 0.31 | 0.06 | 0.35 | 0.02 | |
| | 0.246 | 0.018 | 0.304 | 0.005 | Burton et al. (2015) |
| | 0.26 | 0.03 | 0.27 | 0.01 | Groß et al. (2015) |
| | 0.26 | 0.02 | 0.30 | 0.01 | |
| | 0.252 | 0.03 | 0.28 | 0.02 | Haarig et al. (2017a) |
| Mixed dust | 0.24 | 0.02 | 0.28 | 0.01 | Groß et al. (2011a) |
| | 0.22 | 0.01 | 0.29 | 0.01 | |
| Marine | 0.02 | 0.01 | 0.02 | 0.02 | Groß et al. (2011a) |
| | 0.02 | 0.01 | 0.02 | 0.02 | |
| | 0.05 | 0.01 | 0.07 | 0.01 | Groß et al. (2011b) |
| | 0.04 | 0.01 | 0.05 | 0.01 | Groß et al. (2015) |
| | 0.12 | 0.08 | 0.15 | 0.03 | Haarig et al. (2017b) |
| | 0069 | 0.161 | 0.079 | 0.036 | |
| Mixed anthropogenic | 0.02 | 0.01 | 0.03 | 0.01 | Hoffer et al. (2020) |
| | 0.09 | 0.05 | 0.17 | 0.07 | |
| | 0.14 | 0.05 | 0.23 | 0.06 | |
| | 0.10 | 0.04 | 0.19 | 0.07 | |
| | 0.05 | 0.02 | 0.09 | 0.05 | |

**Table 1.** $\delta_{linear}^{part}$ **at 355 nm and 532 nm (with the corresponding standard deviation) obtained from the literature for dust, marine and mixed anthropogenic aerosol types.**



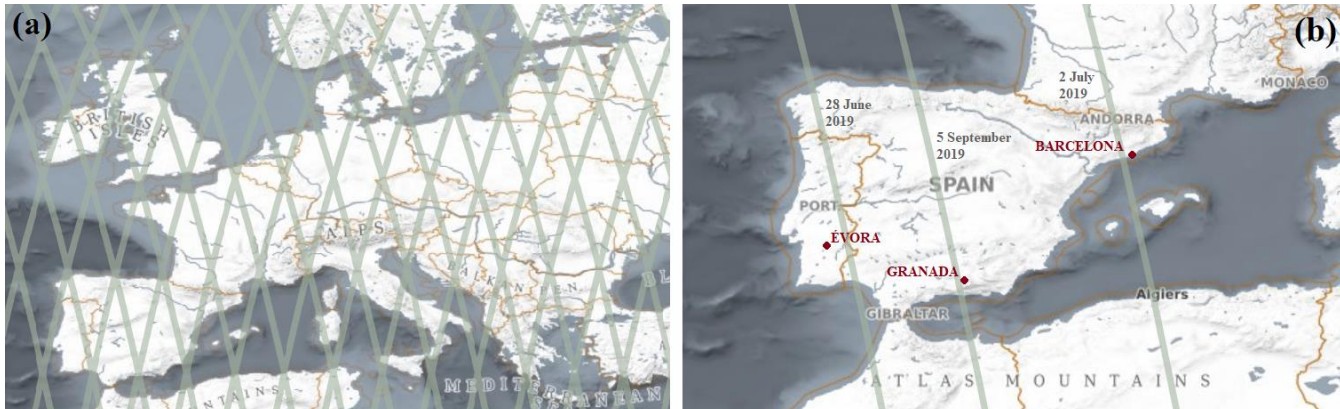

**Figure 1. (a) Distribution of Aeolus overpasses over Europe. (b) Location of the stations in Évora, Granada and Barcelona and the associated overpasses during the case studies analyzed in Section 4.2.. Source: ESA Aeolus online dissemination (aeolus-ds.eo.esa.int).**






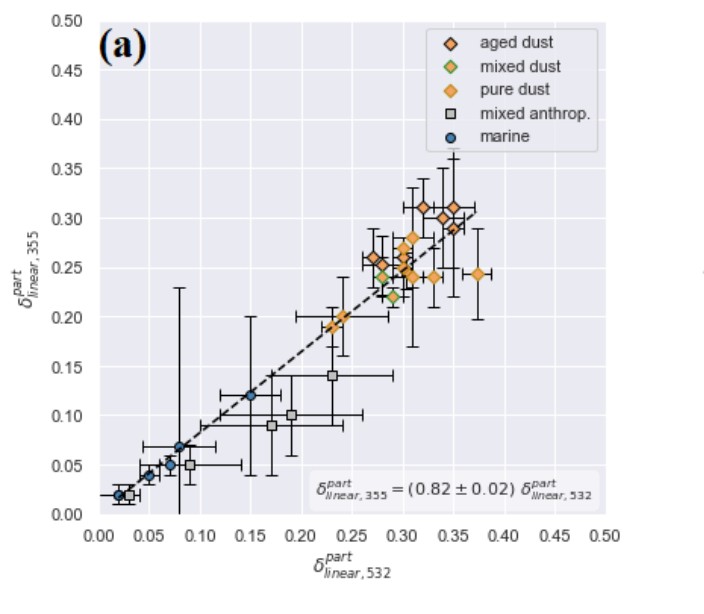

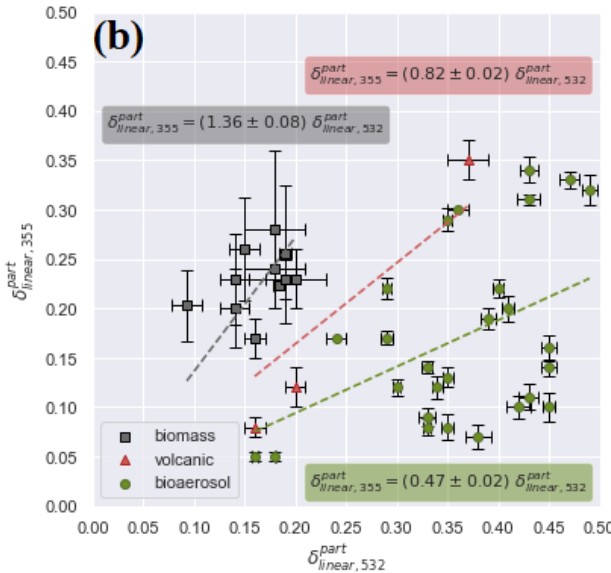

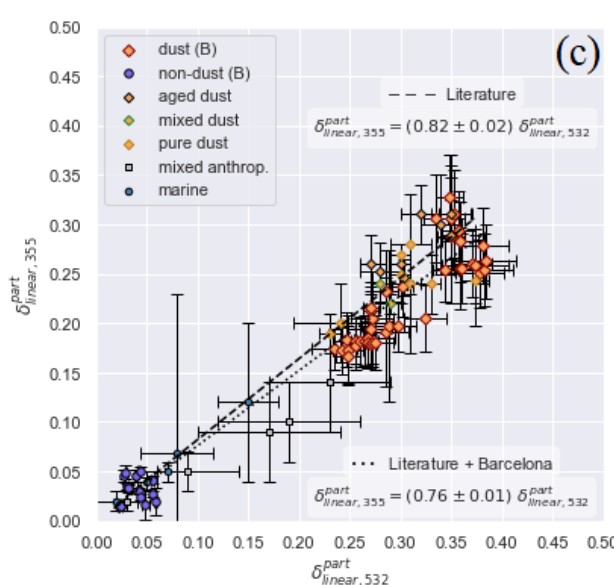

**Figure 2.** Scatter plot of $\delta^{part}_{linear,355}$ versus $\delta^{part}_{linear,532}$ (a) for dust, marine and anthropogenic particles obtained from the literature (see Table 1); (b) for biomass, volcanic and bioaerosol particles; and (c) for dust, marine, anthropogenic particles obtained from the literature and dust and non-dust aerosol particles obtained from dual-polarization measurements in Barcelona. The values for smoke are taken from Groß et al. (2011a), Burton et al. (2015), Haarig et al. (2018), Hu et al. (2018; 2019) and Ohneiser et al. (2020); for volcanic particles from Groß et al. (2012) and for bioaerosols from Cao et al. (2010) and Shang et al. (2020). In Figure 2c Barcelona data are indicated with (B) in the legend.


**Figure 3. (a) Daily range corrected signal measured at 1064 nm in Granada on the 5th September 2019. (b) Aeolus SCAmb backscatter retrievals along the considered orbit (6007) with the profile closest to the station marked in red (source: VirES for Aeolus, aeolus.services). Quality flags are applied.**

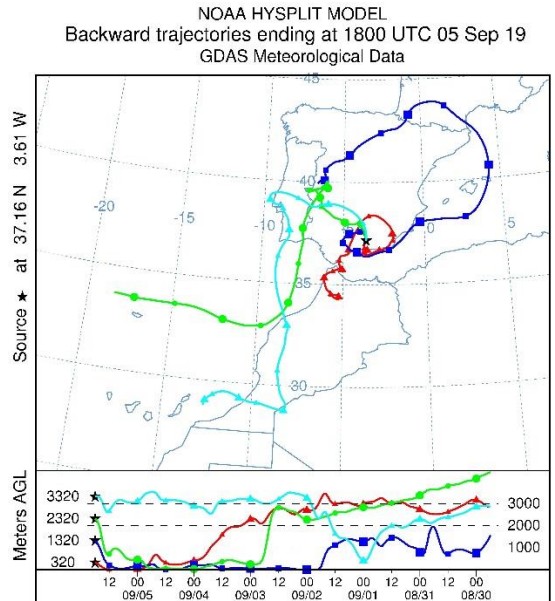

**Figure 4. HYSPLIT model back trajectories for the air masses over Granada between 0.3 and 3.3 km agl (equivalent to 1 and 4 km asl) at 18:00 UTC on the 5th September 2019.**

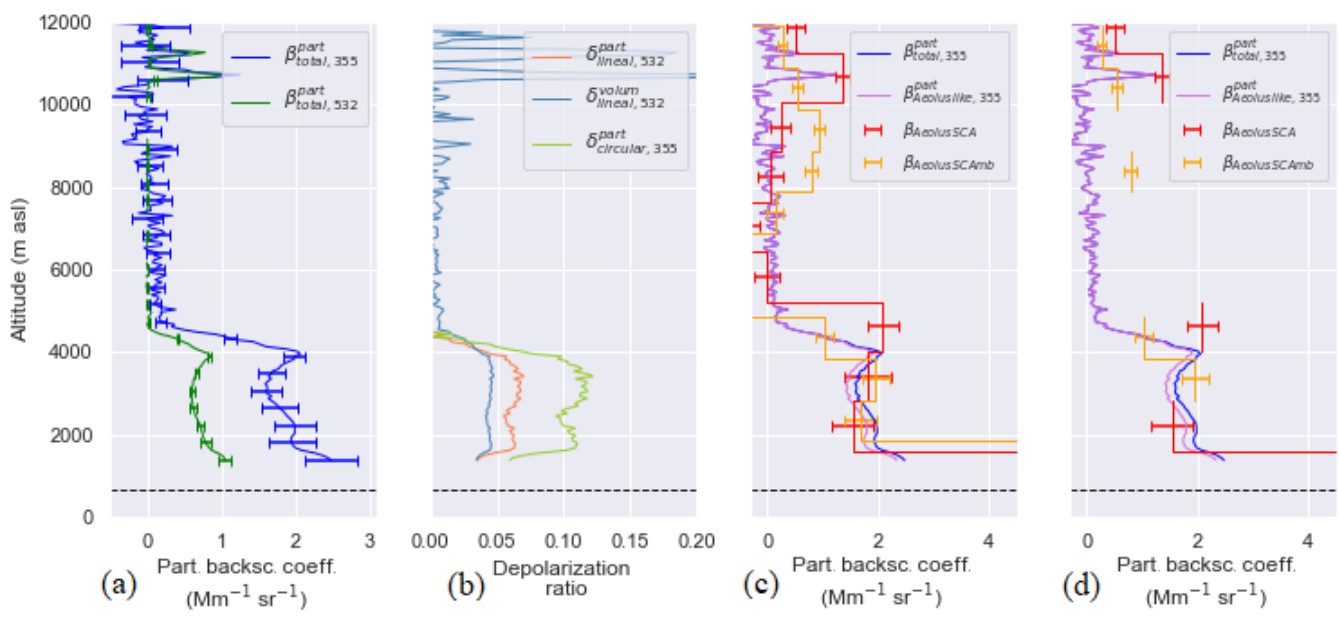

**Figure 5. Vertical profiles for the case study in Granada on the 5th September 2019. (a) Ground-based particle backscatter coefficient profiles at 355 and 532 nm with their uncertainties. (b) Ground-based volume and linear particle depolarization ratios at 532 nm and derived circular particle depolarization ratio at 355 nm. (c) Aeolus SCA and SCAmb co-polar particle backscatter coefficients (without quality flags) and the corresponding ground-based Aeolus-like backscatter coefficient. (d) The same as (c) but considering preliminary quality flags. Ground-based lidar profiles were obtained from the continuous measurement of the system from 18:00 to 18:30 UTC. Satellite-based profiles correspond to the Aeolus overpass at 18:04 UTC.**



**Figure 6. (a) Daily range corrected signal measured with the 355 nm channel in Évora on the 28th June 2019. (b) Aeolus SCAmb backscatter retrievals along the considered orbit (4913) with the profile closest to the station marked in red (source: VirES for Aeolus, aeolus.services.). Quality flags are applied.**

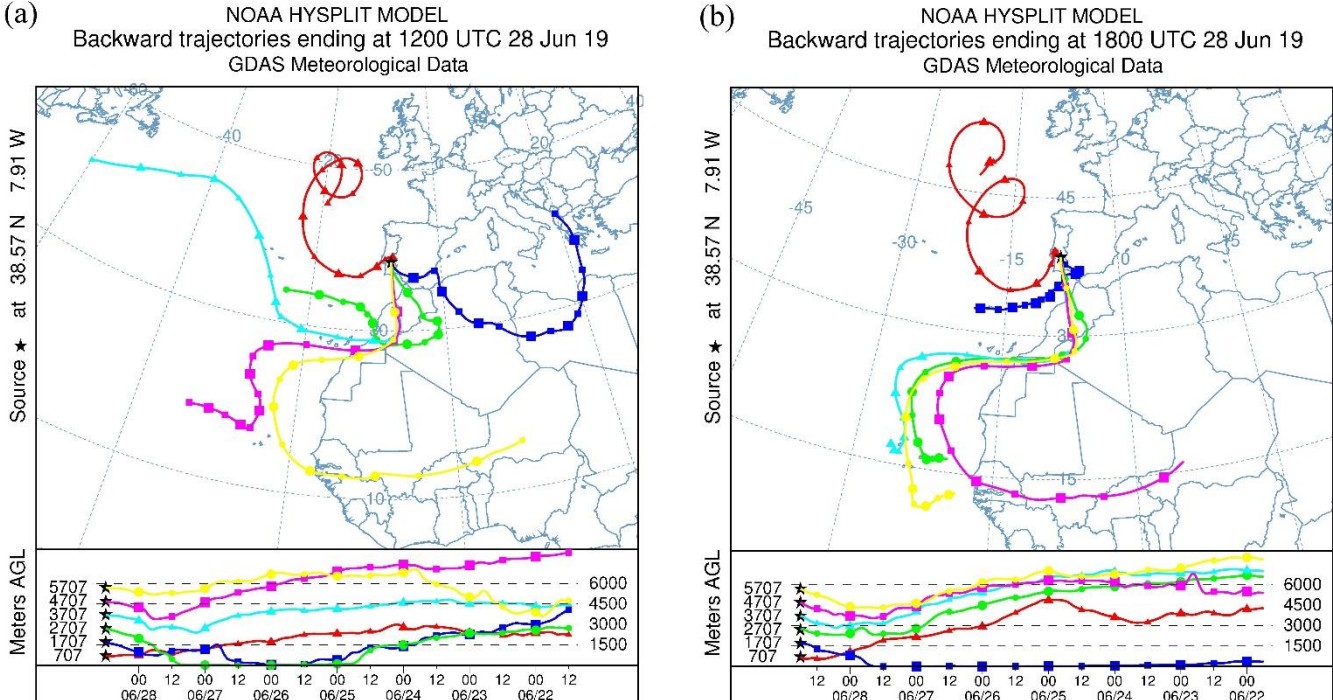

**Figure 7.** HYSPLIT model back trajectories for the air masses over Évora between 0.7 and 5.7 km agl (equivalent to 1 and 6 km asl) on the 28th June 2019 at: (a) 12:00 UTC, (b) 18:00 UTC.

1110

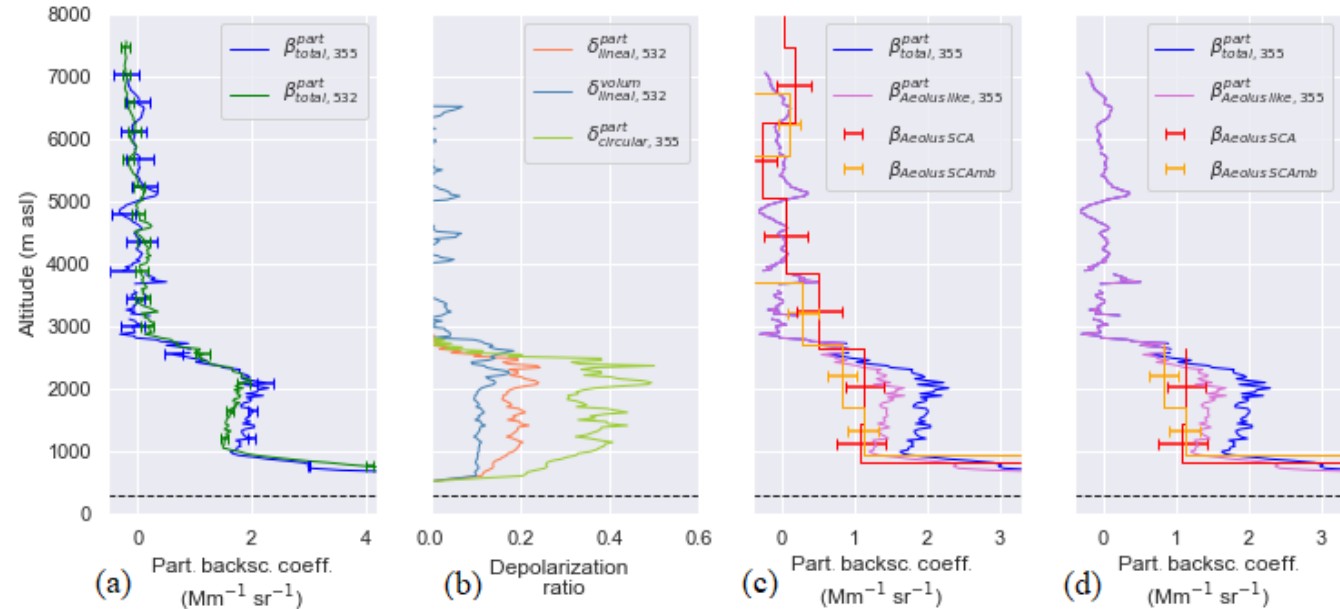

**Figure 8.** Vertical profiles for the case study in Évora on the 28th June 2019. (a) Ground-based particle backscatter coefficient profiles at 355 and 532 nm with their uncertainties. (b) Ground-based volume and linear particle depolarization ratios at 532 nm and derived circular particle depolarization ratio at 355 nm. (c) Aeolus SCA and SCAmb co-polar particle backscatter coefficients (without quality flags) and the corresponding Aeolus-like ground-based backscatter coefficient. (d) The same as (c) but considering

1115





preliminary quality flags. Ground-based lidar profiles were obtained from the continuous measurement of the system from 17:30 to 19:00 UTC. Satellite-based profiles correspond to the Aeolus overpass at 18:17 UTC.

**Figure 9. (a) Daily range corrected signal measured with the 1064 nm channel in Barcelona on the 2nd July 2019. (b) Aeolus SCAmb backscatter retrievals along the considered orbit (4976), with the profile closest to the station marked in red (source: VirES for Aeolus, aeolus.services). Quality flags are applied.**





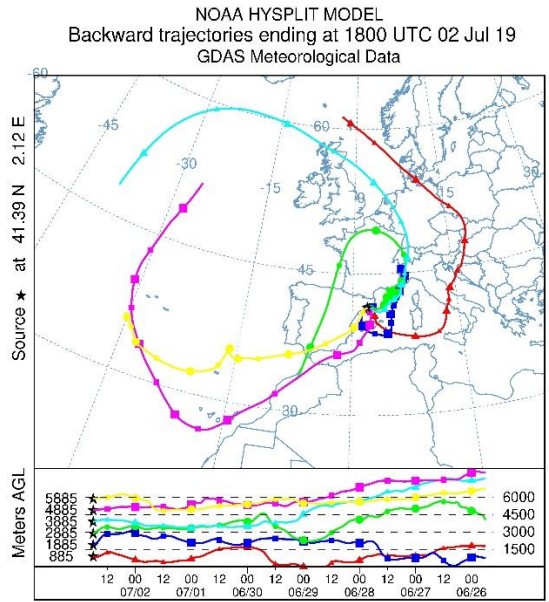

Figure 10. HYSPLIT model back trajectories for the air masses between 0.9 and 5.9 km agl (equivalent to 1 and 6 km asl) over Barcelona at 18:00 UTC on the 2nd July 2019.

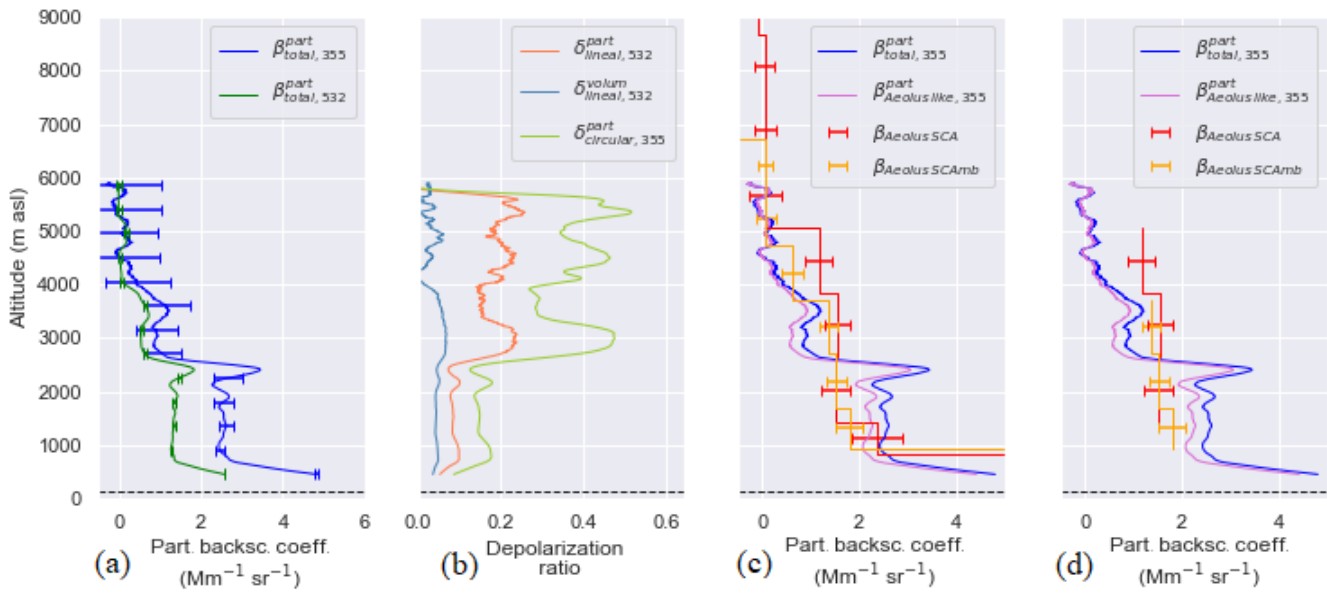

Figure 11. Vertical profiles for the case study in Barcelona on the 2nd July 2019. (a) Ground-based particle backscatter coefficient profiles at 355 and 532 nm with their uncertainties. (b) Ground-based volume and linear particle depolarization ratios at 532 nm and derived circular particle depolarization ratio at 355 nm. (c) Aeolus SCA and SCAmb co-polar particle backscatter coefficients (without quality flags) and the corresponding ground-based Aeolus-like backscatter coefficient. (d) The same as (c) but considering preliminary quality flags. Ground-based lidar profiles were obtained from the continuous measurement of the system from 17:09 to 18:09 UTC. Satellite-based profiles correspond to the Aeolus overpass at 17:39 UTC.





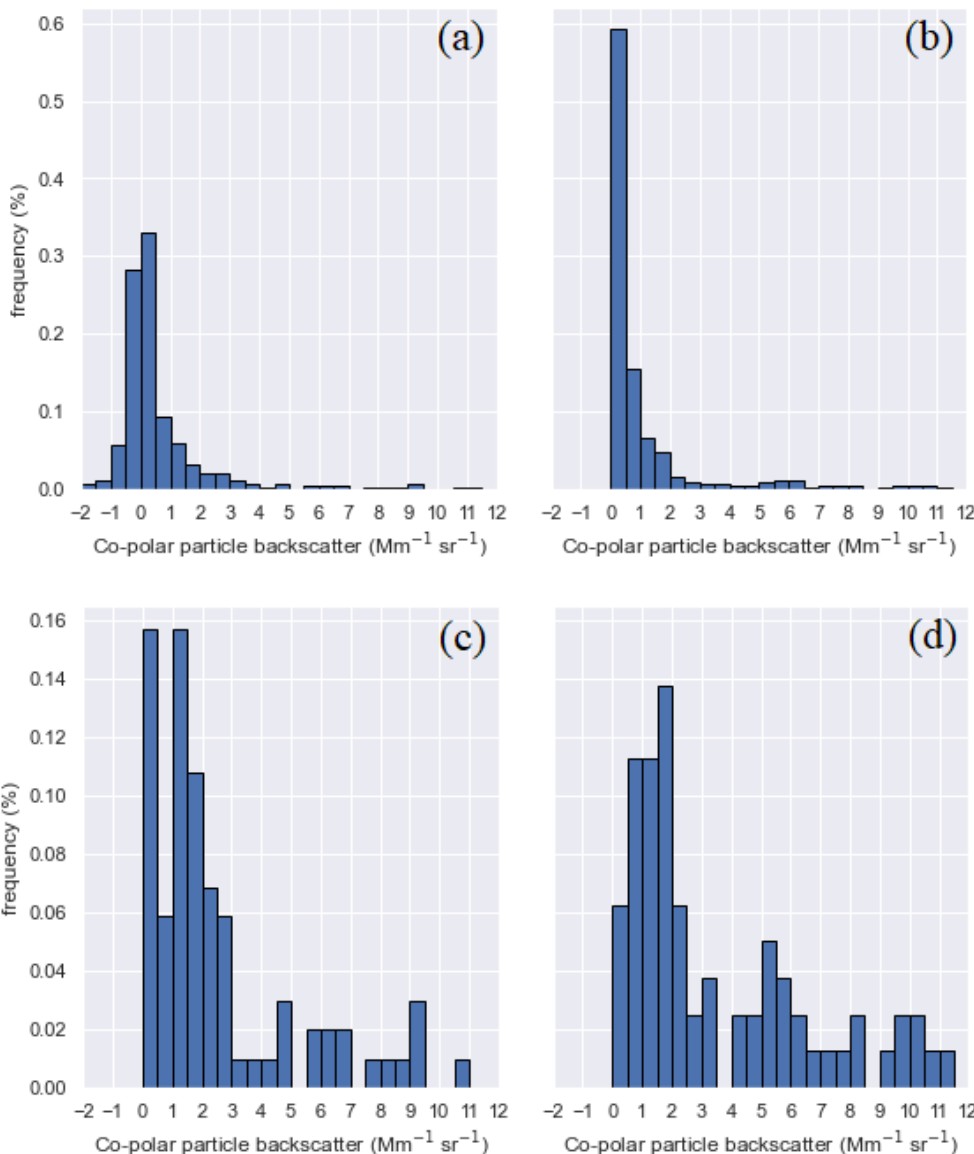

**Figure 12. (a) Aeolus SCA co-polar backscatter coefficient retrievals without the implementation of quality flags for the combined dataset of the considered overpasses at the three stations. (b) Aeolus SCAmb co-polar backscatter coefficient retrievals without the implementation of quality flags for the combined dataset of the considered overpasses at the three stations. (c) same as (a) but considering quality flags. (d) same as (b) but considering quality flags.**

1140





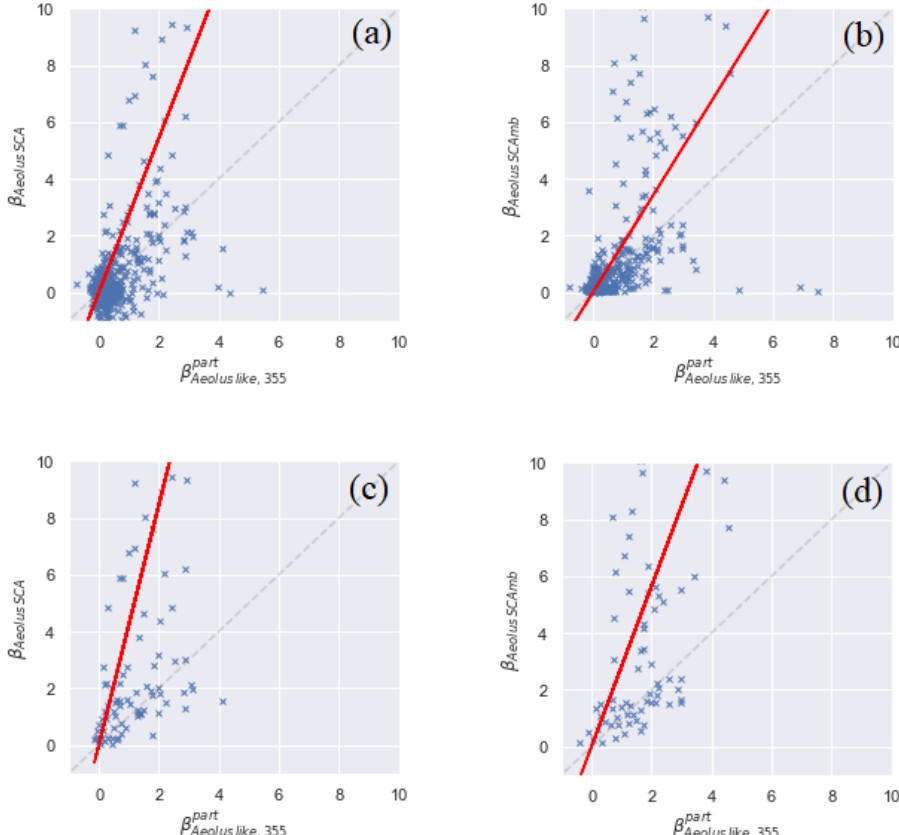

**Figure 13.** (a) $\beta^{part}_{Aeolus\ like,355}$ and $\beta_{Aeolus\ SCA}$ of the combined database with no quality flags applied. (b) $\beta^{part}_{Aeolus\ like,355}$ and $\beta_{Aeolus\ SCAmb}$ of the combined database with no quality flags applied. (c) same as (a) but considering quality flags. (d) same as (b) but considering quality flags. The values of each dataset have been adjusted to a linear model with null intercept (red line).

1145





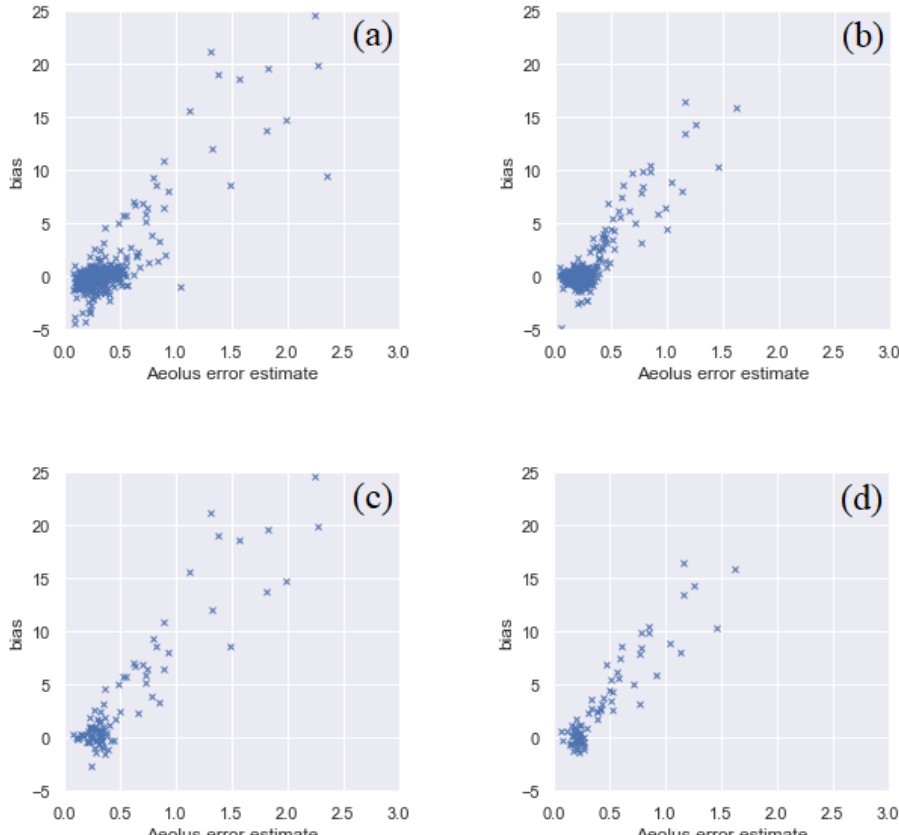

**Figure 14. (a) Bias and Aeolus error estimate of the SCA products of the whole database with no quality flags applied, (b) bias and Aeolus error estimate of the SCAmb products of the whole database with no quality flags applied, (c) same as (a) but considering quality flags, (d) same as (b) but considering quality flags.**



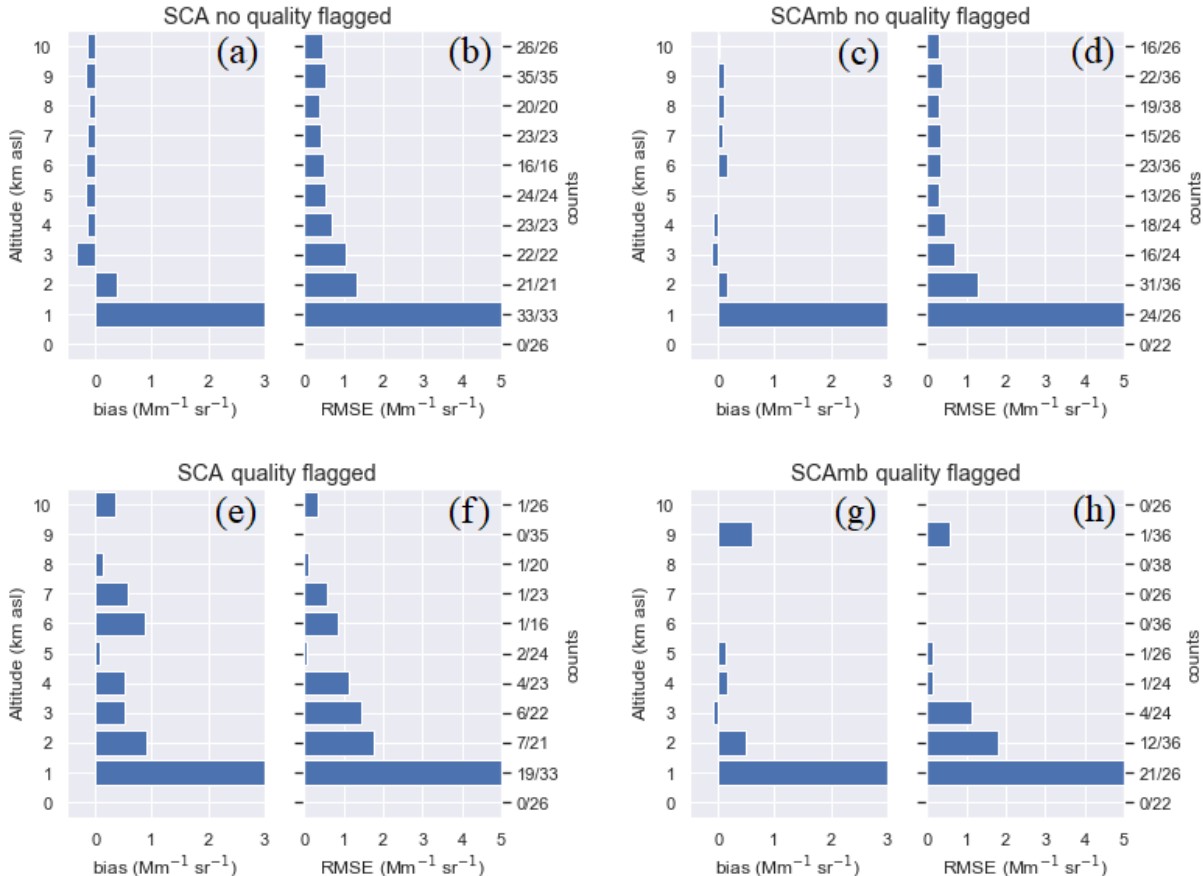

**Figure 15. Statistical results for the intercomparison of Aeolus SCA and SCAmb products with Granada ground-based measurements, with and without quality flags. The right-hand axis indicates the number of available data points included in each vertical range out of the total number of measures within that vertical range. (a) bias and (b) RMSE of the intercomparison of the SCA products without quality flags applied. (c) bias and (d) RMSE of the intercomparison of the SCAmb products without quality flags applied. (e) and (f) same as (a) and (b) but considering quality flags. (g) and (h) same as (c) and (d) but considering quality flags.**

1155



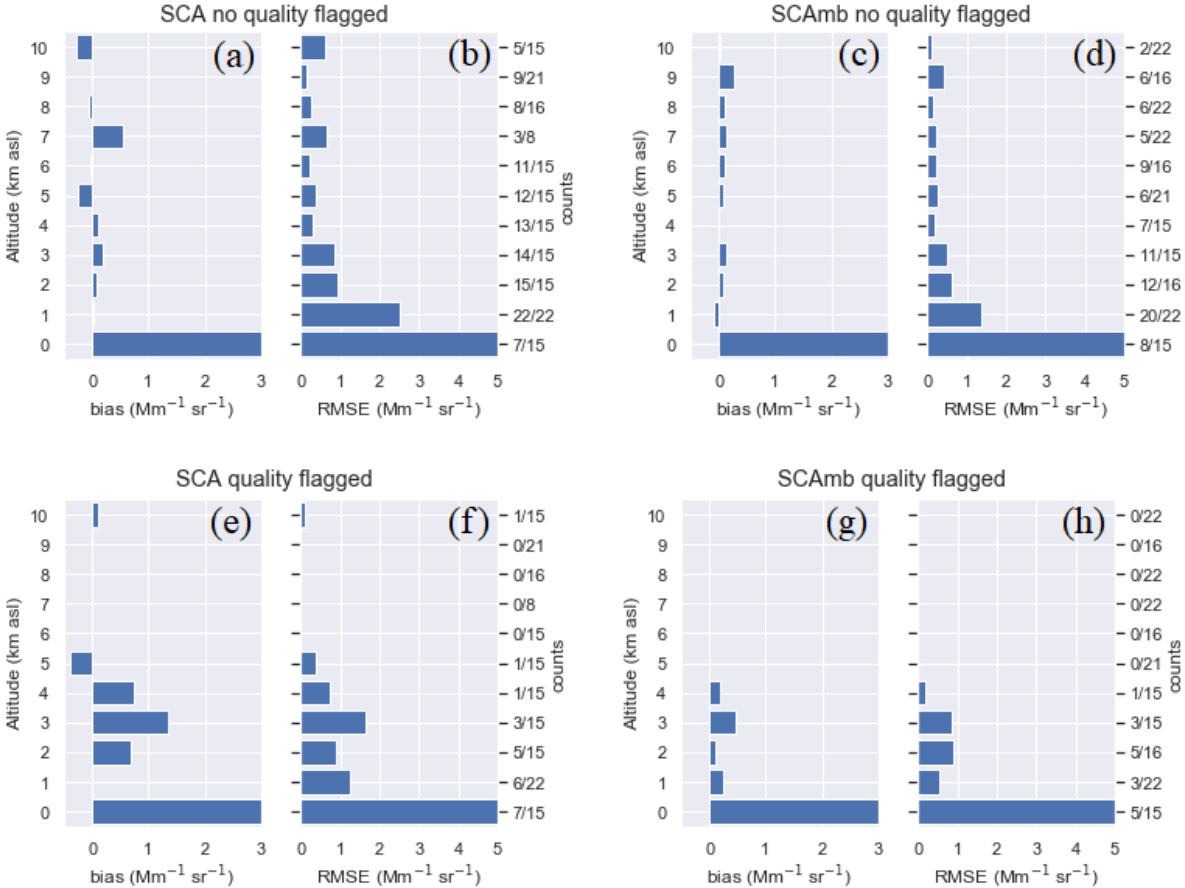

**Figure 16. Statistical results for the intercomparison of Aeolus SCA and SCAmb products with Granada ground-based measurements, with and without quality flags. The right-hand axis indicates the number of available data points included in each vertical range out of the total number of measures within that vertical range. (a) bias and (b) RMSE of the intercomparison of the SCA products without quality flags applied. (c) bias and (d) RMSE of the intercomparison of the SCAmb products without quality flags applied. (e) and (f) same as (a) and (b) but considering quality flags. (g) and (h) same as (c) and (d) but considering quality flags.**



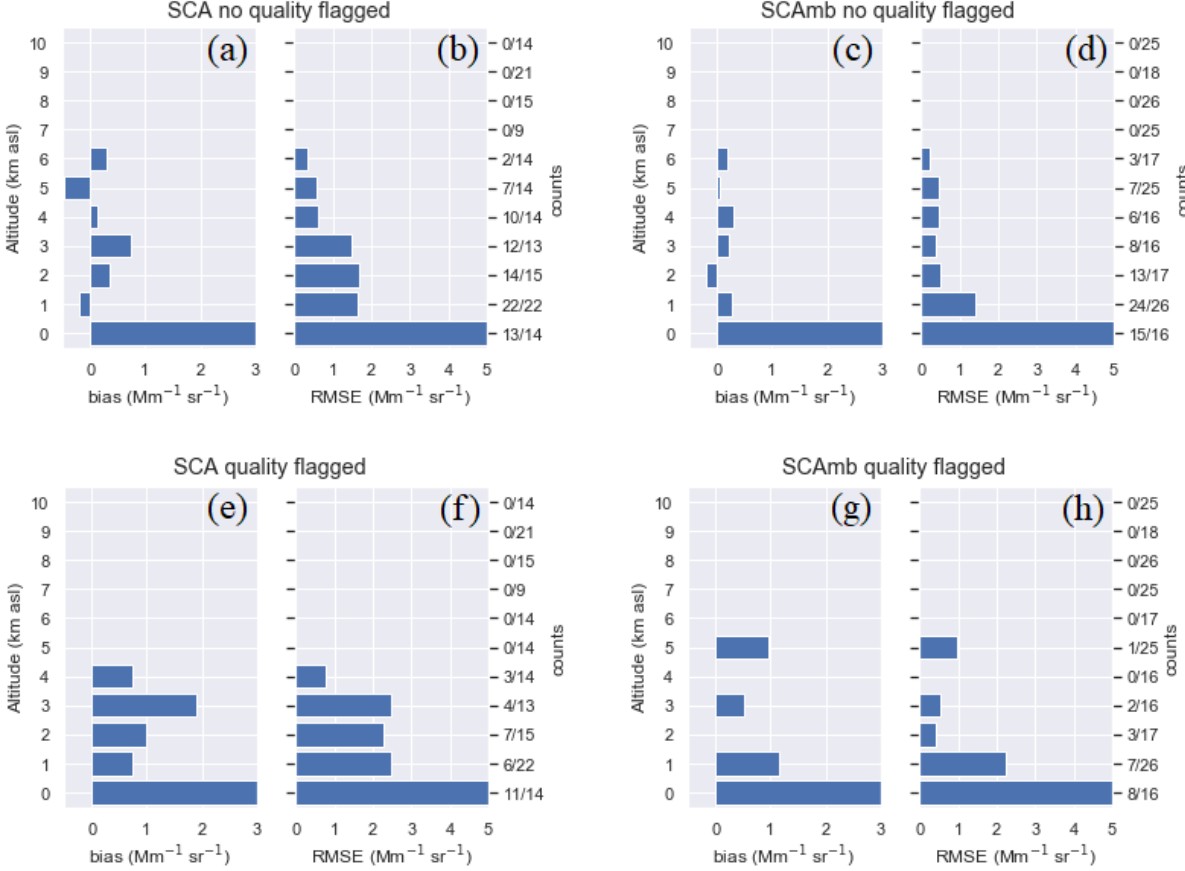

**Figure 17. Statistical results for the intercomparison of Aeolus SCA and SCAmb products with Barcelona ground-based measurements, with and without quality flags. The right-hand axis indicates the number of available data points included in each vertical range out of the total number of measures within that vertical range. a) bias and (b) RMSE of the intercomparison of the SCA products without quality flags applied. (c) bias and (d) RMSE of the intercomparison of the SCAmb products without quality flags applied. (e) and (f) same as (a) and (b) but considering quality flags. (g) and (h) same as (c) and (d) but considering quality flags.**