# Peer review of "Statistical validation of Aeolus L2A particle backscatter coefficient retrievals over ACTRIS/EARLINET stations in the Iberian Peninsula"

_Atmospheric Chemistry and Physics, 2021_

## Referee Comment (RC1)

**Statistical validation of Aeolus L2A particle backscatter coefficient retrievals over ACTRIS/EARLINET stations in the Iberian Peninsula**

Jesús Abril-Gago, Juan Luis Guerrero-Rascado, Maria João Costa, Juan Antonio Bravo-Aranda, Michaël Sicard, Diego Bermejo-Pantaleón, Daniele Bortoli, María José Granados-Muñoz, Alejandro Rodríguez-Gómez, Constantino Muñoz-Porcar, Adolfo Comerón, Pablo Ortiz-Amezcua, Vanda Salgueiro, Marta María Jiménez-Martín, and Lucas Alados-Arboledas

Comments on Preprint acp-2021-388

In this manuscript the authors provide a good introduction to the uncertainties regarding the measurements of aerosols present in the atmosphere and the interaction between aerosols and clouds, mentioning the importance of lidar networks and measurement campaigns to reduce the aerosol optical properties uncertainties and improve data quality for satellite data calibration. The authors present a highly valuable data set based on lidar measurements for three different EARLINET lidar station to assess the Aeolus satellite reprocessed data (baseline 10). Since the Aeolus satellite receiver detects the co-polar component of circular polarized backscattered radiation at 355 nm, the authors applied a technique to extract the co-polar component of the ground-based observations at 355 nm from the total particle backscatter coefficient. In the manuscript the authors present three different case studies and a statistical analysis of the baseline 10 co-polar backscatter coefficients from the standard correct algorithm middle bin (SCAmb) and the standard correct algorithm (SCA). The manuscript is well written and the results are very important to validate the aerosol detection of Aeolus mission. I recommend the article for publication following correction and clarification of a few minor issues described below.

In the section 3.1-Database and intercomparison methodology (page 7) the authors describe the measurement protocol for each station - For Granada, a 1.5 hour interval for the morning overpass time and a 1-hour interval for the evening overpass time (i.e.17:30 - 18:30 UTC) were chosen. For Barcelona, a 1-hour range centered at the overpass time was considered. For Évora, a 1.5-hour interval containing the overpass time was considered to take into account the larger distance between the Aeolus ground track and the lidar site. I recommend the authors discuss in more details the uncertainties or issues that 1 or 1.5-hour time differences can cause in the comparative analyzes between the ground-based lidar and the Aeolus satellite.

In section 3.2-Aeolus-like conversion of ground-based lidar particle backscatter coefficients - The authors proposed a method to estimate the linear particle depolarization ratio at 355 nm ($\delta^{part}_{linear,355}$) from the linear particle depolarization ratio at 532 nm. However, it is stated the third lidar station in Barcelona, does measure both depolarization ratios but for the sake of consistency of the data processing, calculated the same way than the other two stations. In figure 2c (page 39 and discussion on page 10) the autors presented a scatter plot of dust and non-dust aerosol particles obtained from dual-polarization measurements in Barcelona, together with spectral conversion factor Kδ=0,76±0,01 and the spectral conversion factor from literature results for dust and non-dust types, equals to 0.82 ± 0.02. It is not very

clear why the authors prefer to use the literature values instead of spectral conversion factor retrieved from measurements.

In section 3.3 Statistical parameters, the authors stated *"The resolution of these bins depends on the altitude range: 500 m between 0 and 2 km asl, 1 km between 2 and 16 km asl and 2 km between 16 and 30 km asl. Because the ground-based lidars present a much finer resolution, of the order of a few meters, the resolution of each ground -based profile has been degraded to the Aeolus vertical resolution."*
The authors' choice to downgrade the data quality of the groundbased handles to perform a bin-to-bin comparison is understandable, however, the decrease in signal quality doesn't seem to make much sense when comparing the Aeolus and the groundbased lidar, especially when taking into account the different nuances of the atmosphere in the region closest to the surface. How do the authors understand that this loss of quality, or the lack of a finer resolution in the Aeolus data, can affect the application of the data to the study of optical properties of aerosols?

In section 4.3 Case studies, the authors stated *"Sun -photometer measurements are taken into account for the sake of completeness aerosol typing, through the study of the aerosol optical depth at 675 nm (AOD 675 )"*. Why was the 657 nm wavelength chosen? Why not choose the AOD values in the UV region as 340 or 380 nm, instead?

*"The location of the stations is highly interesting due to their proximity to the Sahara Desert and mainland Europe, so frequent events of mineral dust and anthropogenic particles could be detected by the satellite. In addition, Barcelona lies just in the coastline, and both Barcelona and Granada present high concentrations of anthropogenic aerosol, while Évora aerosol concentrations could be classified as rural. Thus, Aeolus operation can be tested under a complete set of atmospheric scenarios."*
How was the difficulty of comparing the layers closest to the surface taken into account that Barcelona station is located just in the coastline and is influenced by the mixture of anthropogenic aerosol and/or dust and marine aerosol? How might this difficulty in comparing the layers closest to the surfacehave influenced the statistical results?

In Page 13-lines 385-386 - *"The HYSPLIT model indicates that the 12:00 UTC air masses over Évora at 1.7 and 2.7 km agl (equivalent to 2 and 3 km asl) are coming directly from lower altitudes in Northern Africa (Figure 7a)."*
This sentence is slightly confused, please, rewrite the sentence.

Page 15 - Lines 455 and 456 - *"First, the satellite presents a satisfactory agreement with the ground-based lidar in the whole available profile under both SCA and SCAmb (Figure 11c)."*
For the first atmospheric layers up to 2,5 km asl it seems there is an underestimation of Aeolus particle backscatter signal, and for the layers from 2.5 to 6 km asl, it seems to have an overestimation of Aeolus particle backscatter signal. Considering that Barcelona is the station with the most complex scenario, with several layers coming from different sources and containing different optical properties, the comparison analysis seems to be much more sensitive, I believe the authors could explore more this aspects in order to improve the manuscript discussion.

Page 15 - lines 467-469 - *"101 B10-overpasses for Granada, 51 for Évora and 52 for Barcelona, and after applying the set of requirements, the intercomparison has been performed with 24 cases for Granada, 15 cases for Évora and 16 cases for Barcelona, leading to enough statistical significance."*
What criteria were considered by the authors to reach the conclusion that this number of cases is statistically significant?

Page 15 - lines 474-475 - *"With the implementation of the quality flags (Figure 12c and 12 d), all of the sets range from 0 $Mm^{-1}sr^{-1}$."*.
Please, consider correct this sentence. All the sets range from 0 to which value?

Page 16 - lines 490-491 - *"Aeolus backscatter coefficient uncertainties (known as Aeolus error estimates) are addressed through the biases between satellite and ground-based measurements. Figure 14 reveals that the larger the Aeolus uncertainties, the larger the bias."*
Just for improve the understanding, is the bias mentioned in this sentence and presented in axis Y in figure 14 the same values calculated in equation presented in page 9-line 286? If yes, I would recomment the authors rewrite the sentence.

Page 23 - line 710 - Please, consider correct the reference *"Córdoba-Jabonero, C., Sicard, M., López-Cayuela, M.-A., Ansmann, A., Comerón, A., Zorzano, M.-P., Rodríguez-Gómez, A., and Muñoz-Porcar, C.: Aerosol radiative effect during the summer 2019 heatwave produced partly by an intercontinental Saharan dust outbreak. 1. Shortwave dust-induced direct impact, Atmospheric Chemistry and Physics, 21, 1–25, https://doi.org/10.5194/acp-2020-1013, 2021."* since the DOI is leading to the pre-printed version of the article.

---

## Author Comment (AC1)

Dear reviewers and editor:

We would like to thank you for taking the time to review our manuscript and suggest such valuable information in order to improve its quality. Even the most trivial suggestion helped in the progress of the text's value.

Some main and relevant changes have been made to the manuscript, as summarized below:

- While submitting the first version of our manuscript, a relevant study related to Aeolus optical products were made. Thus, for the sake of completeness, the following published paper has been included now in the reviewed version of our work:

  Baars, H., Radenz, M., Floutsi, A. A., Engelmann, R., Althausen, D., Heese, B., Ansmann, A., Flament, T., Dabas, A., Trapon, D., Reitebuch, O., Bley, S., & Wandinger, U. (2021). Californian Wildfire Smoke Over Europe: A First Example of the Aerosol Observing Capabilities of Aeolus Compared to Ground-Based Lidar. Geophysical Research Letters, 48(8), e2020GL092194. https://doi.org/10.1029/2020GL092194

- Furthermore, a reprocessing of Granada's ground-based dataset was made right after the manuscript submission, to increase the quality of the depolarization-related products. No relevant differences were found between the results obtained with the former and the reprocessed products. However, the charts included in the manuscript have been updated accordingly. These changes affect Figures 5, 12, 13, 14 and 15 (a new figure is included following the Reviewer#2 suggestion, so these figures correspond now to Figure 5, 13, 14, 15, and 16, respectively).

Apart from these main changes, a set of minor changes and typos has been corrected throughout the document. Following this introduction, you may find a detailed answer to every Reviewer#1's comment. All of the listed changes and the ones suggested by the reviewers can be seen in the new version of the manuscript, marked in red.

**Reviewer 1**

1) *In the section 3.1-Database and intercomparison methodology (page 7) the authors describe the measurement protocol for each station - For Granada, a 1.5 hour interval for the morning overpass time and a 1-hour interval for the evening overpass time (i.e.17:30 - 18:30 UTC) were chosen. For Barcelona, a 1-hour range centered at the overpass time was considered. For Évora, a 1.5-hour interval containing the overpass time was considered to take into account the larger distance between the Aeolus ground track and the lidar site. I recommend the authors discuss in more details the uncertainties or issues that 1 or 1.5-hour time differences can cause in the comparative analyzes between the ground-based lidar and the Aeolus satellite.*

Each station is managed by independent research groups, which means that some minor differences can be found in the processing of the products, although all of them are processed according to the EARLINET protocols, processing chain and standards. In particular, each group produces its products under different temporal periods (i.e. 30 minutes or 1 hour). This fact, plus the need of an adequately long averaged period of measurements, made us use different temporal averages for each station. Regarding the temporal evolution of the meteorological conditions, it has been checked that during the 1 hour or 1.5 hour interval the atmospheric conditions had not changed significantly, so that the temporal averaging of the signals is coherent in the interval. Thus, we get to choose the most convenient temporal average for each station as no uncertainties nor artifacts are introduced from the fact that two sets of averages have been chosen. Thus, the next clarification has been added in the new version of the manuscript:

The temporal evolution of the meteorological conditions and layers has been checked individually, so as to ensure that the selection of 1 hour or 1.5 hour as average interval for the lidar signals does not introduce any artifact in the comparison.

2) *In section 3.2-Aeolus-like conversion of ground-based lidar particle backscatter coefficients - The authors proposed a method to estimate the linear particle depolarization ratio at 355 nm ($\delta^{part}_{linear, 355}$) from the linear particle depolarization ratio at 532 nm. However, it is stated the third lidar station in Barcelona, does measure both depolarization ratios but for the sake of consistency of the data processing, calculated the same way than the other two stations. In figure 2c (page 39 and discussion on page 10) the autors presented a scatter plot of dust and non-dust aerosol particles obtained from dual-polarization measurements in Barcelona, together with spectral conversion factor Kδ=0,76±0,01 and the spectral conversion factor from literature results for dust and non-dust types, equals to 0.82 ± 0.02. It is not very clear why the authors prefer to use the literature values instead of spectral conversion factor retrieved from measurements.*

On one hand, we set up a state-of-the-art database of dual-polarization measurements covering all the aerosol types reported in the literature up to the date, offering a complete insight worldwide.

On the other hand, Barcelona station provides independent measurements that undergo a set of validation criteria but have not been published in any peer-reviewed journal yet. Moreover, the Barcelona depolarization ratios measurements at 355 nm do not cover the complete period analyzed in our study, and are mainly biased to dust cases.

Considering all these arguments, we preferred to use the spectral conversion factor derived from the literature in order to base our results on measurements of different stations (i.e. different environmental conditions and aerosol types), which themselves have been reviewed through peer review process. Thus, we included the following explanation in the mentioned paragraph:

The literature-derived factor is implemented in order to base the results in previous measurements reported for different environmental conditions and aerosol types, which are subsequently used as reference in other studies.

3) *In section 3.3 Statistical parameters, the authors stated "The resolution of these bins depends on the altitude range: 500 m between 0 and 2 km asl, 1 km between 2 and 16 km asl and 2 km between 16 and 30 km asl. Because the ground-based lidars present a much finer resolution, of the order of a few meters, the resolution of each ground -based profile has been degraded*

*to the Aeolus vertical resolution." The authors' choice to downgrade the data quality of the groundbased handles to perform a bin-to-bin comparison is understandable, however, the decrease in signal quality doesn't seem to make much sense when comparing the Aeolus and the groundbased lidar, especially when taking into account the different nuances of the atmosphere in the region closest to the surface. How do the authors understand that this loss of quality, or the lack of a finer resolution in the Aeolus data, can affect the application of the data to the study of optical properties of aerosols?*

This question is particularly interesting because of several facts. On the one hand, the vertical resolution of Aeolus profiles for aerosol products is really coarse compared to the vertical resolution of other satellites, like CALIPSO. Thus, the readers have to bear in mind that Aeolus vertical resolution does not provide a detailed characterization of the atmospheric optical properties. Aeolus products do provide valuable information for the detection of significant layers and clouds, as it can clearly be seen in the case studies provided (Section 4.2). On the other hand, the atmospheric layer with more relevant nuances happens to be in the lowermost troposphere, where Aeolus is proven to fail (at the very bottom). Therefore, once again, we can say that Aeolus provides valuable information for the characterization of significant layers and clouds. Thus, the following explanation is now included in the conclusions section:

However, as it can be noted from the results, Aeolus vertical resolution is too coarse (especially compared to other satellites) for a detailed characterization of the nuances of the atmospheric optical properties. Thus, Aeolus provides valuable information in the detection and characterization of significant aerosol and cloud layers.

4) *In section 4.3 Case studies, the authors stated "Sun -photometer measurements are taken into account for the sake of completeness aerosol typing, through the study of the aerosol optical depth at 675 nm (AOD 675 )". Why was the 657 nm wavelength chosen? Why not choose the AOD values in the UV region as 340 or 380 nm, instead?*

The 675 nm wavelength was chosen (among all of the Sunphotometer wavelengths) as it is one of the wavelengths most used as reference by the scientific community. Furthermore, Barcelona's Sunphotometer only worked with 440, 500, 675, 870 and 1020 nm on the 2nd July 2019, so the 340 or 380 nm channels could not be used. Thus, for the sake of homogeneity, the 675 nm channel was chosen as a common reference for the three stations.

5) *"The location of the stations is highly interesting due to their proximity to the Sahara Desert and mainland Europe, so frequent events of mineral dust and anthropogenic particles could be detected by the satellite. In addition, Barcelona lies just in the coastline, and both Barcelona and Granada present high concentrations of anthropogenic aerosol, while Évora aerosol concentrations could be classified as rural. Thus, Aeolus operation can be tested under a complete set of atmospheric scenarios." How was the difficulty of comparing the layers closest to the surface taken into account that Barcelona station is located just in the coastline and is influenced by the mixture of anthropogenic aerosol and/or dust and marine aerosol? How might this difficulty in comparing the layers closest to the surface have influenced the statistical results?*

With the quoted paragraph we attempted to express that we tried to assess Aeolus performance under different situations although the focus was not on the differences this set of scenarios might produce. Indeed, the objective of this study is to test Aeolus performance, and luckily we are allowed to work

with different scenarios and not just one (i.e. only coastline/marine or flat/rural settings). Thus, the aerosol mixture state does not statistically affect the results. As presented in Section 4.3, the geographical differences of the stations may affect the statistical results, with notorious differences in the lowermost regions, affected by the surface. Consequently, the statistical analysis presented in Section 4.3 was performed independently for each station so as to detect these differences. Furthermore, as previously mentioned in the answer#3, Aeolus vertical resolution is too coarse (when compared to other satellites as CALIPSO) limiting the detailed characterization of the nuances of the atmospheric optical properties at the lowermost atmosphere.

Regarding Barcelona station location just in the coastline, as the satellite overpasses the station at a close distance we can assume that both instruments (ground-based and space-borne lidars) detect the same air masses. Consequently, both instruments will register the same effects that the geographical layout might produce, so no special considerations have to be taken into account in the statistical analysis.

6) *In Page 13-lines 385-386 - "The HYSPLIT model indicates that the 12:00 UTC air masses over Évora at 1.7 and 2.7 km agl (equivalent to 2 and 3 km asl) are coming directly from lower altitudes in Northern Africa (Figure 7a)." This sentence is slightly confused, please, rewrite the sentence.*

The following sentence will be included instead of the quoted one:

The HYSPLIT model indicates that at 12:00 UTC the air mass located over Évora at 1.7 and 2.7 km agl (equivalent to 2 and 3 km asl) originates from surface-level of Northern Africa (Figure 7a).

7) *Page 15 - Lines 455 and 456 - "First, the satellite presents a satisfactory agreement with the ground-based lidar in the whole available profile under both SCA and SCAmb (Figure 11c)." For the first atmospheric layers up to 2,5 km asl it seems there is an underestimation of Aeolus particle backscatter signal, and for the layers from 2.5 to 6 km asl, it seems to have an overestimation of Aeolus particle backscatter signal. Considering that Barcelona is the station with the most complex scenario, with several layers coming from different sources and containing different optical properties, the comparison analysis seems to be much more sensitive. I believe the authors could explore more this aspects in order to improve the manuscript discussion.*

The development of a more detailed comparison analysis for this particular and interesting case study could be really rewarding. Unfortunately, regarding Aeolus limitations it is not possible to increase the detail of the comparison, as we have to work with Aeolus fixed vertical resolution.

Furthermore, as stated in Section 4.1, a single conversion factor $K_\delta$ is considered in the intercomparison, in order to minimize the uncertainties and the effects that different aerosol types might cause. However, hereunder we explore the dependency of the Aeolus-like profile depending on the $K_\delta$, this is, depending on the aerosol types considered in Section 4.1. Thus, a set of $K_\delta$ values have been taken: 0.82, for bibliographic (considering all aerosol types); 0.76 for the whole set of Barcelona cases; 0.72 for the set of dust cases in Barcelona; and 0.90 for the set of non-dust cases in Barcelona. The following plot is a zoom of the results in a way that the differences are somehow visible. We can see slight differences between the profiles for different $K_\delta$ values. In fact, the largest difference can be seen between the $K_\delta = 0.90$ and the $K_\delta = 0.72$ profiles, and they differ only in 3%.

[Figure]

**Figure.** Aeolus SCA and SCAmb co-polar particle backscatter coefficients (without quality flags) and the corresponding ground-based Aeolus-like backscatter coefficient (considering a set of different $K_\delta$ values) for the case study in Barcelona on the 2nd July 2019.

Thus, a more detailed study of the sources and optical properties of the aerosol layer will not improve the intercomparison assessed in the study.

8) *Page 15 - lines 467-469 - "101 B10-overpasses for Granada, 51 for Évora and 52 for Barcelona, and after applying the set of requirements, the intercomparison has been performed with 24 cases for Granada, 15 cases for Évora and 16 cases for Barcelona, leading to enough statistical significance." What criteria were considered by the authors to reach the conclusion that this number of cases is statistically significant?*

We wanted to express that the set of different scenarios considered with the mentioned dataset of cases allows us to test Aeolus performance under different circumstances. No particular criteria is considered to reach the conclusion that the number of cases is statistically significant. Now, the quoted sentence has been restated as:

101 B10-overpasses for Granada, 51 for Évora and 52 for Barcelona were considered, and after applying the set of requirements, the intercomparison has been performed with 21 cases for Granada, 15 cases for Évora and 16 cases for Barcelona, leading to a wide dataset of cases.

9) *Page 15 - lines 474-475 - "With the implementation of the quality flags (Figure 12c and 12 d), all of the sets range from 0 Mm-1sr -1 .". Please, consider correct this sentence. All the sets range from 0 to which value?*

The quoted sentence has been restated and completed as:

With the implementation of the quality flags (Figure 13c and 13d), all of the sets range from 0 Mm$^{-1}$ sr$^{-1}$ onwards. Actually, the maximum values mentioned are still flagged as valid, 86 Mm$^{-1}$ sr$^{-1}$ and 68 Mm$^{-1}$ sr$^{-1}$ in the case of the SCA and SCAmb, respectively.

10) *Page 16 - lines 490-491 - "Aeolus backscatter coefficient uncertainties (known as Aeolus error estimates) are addressed through the biases between satellite and ground-based measurements. Figure 14 reveals that the larger the Aeolus uncertainties, the larger the bias." Just for improve the understanding, is the bias mentioned in this sentence and presented in axis Y in figure 14 the same values calculated in equation presented in page 9-line 286? If yes, I would recommend the authors rewrite the sentence.*

The bias mentioned in lines 490-491 and presented in Figure 14 is indeed the parameter mentioned in line 286. In fact, it is the only bias mentioned in the manuscript. The quoted sentence has been restated and completed as:

Aeolus backscatter coefficient uncertainties (known as Aeolus error estimates) are addressed through the biases between satellite and ground-based measurements (as presented in Section 3.3).

11) *Page 23 - line 710 - Please, consider correct the reference "Córdoba-Jabonero, C., Sicard, M., López-Cayuela, M.-A., Ansmann, A., Comerón, A., Zorzano, M.-P., Rodríguez-Gómez, A., and Muñoz-Porcar, C.: Aerosol radiative effect during the summer 2019 heatwave produced partly by an intercontinental Saharan dust outbreak. 1. Shortwave dust-induced direct impact, Atmospheric Chemistry and Physics, 21, 1–25, https://doi.org/10.5194/acp-2020-1013, 2021." since the DOI is leading to the pre-printed version of the article.*

The quoted citation has been corrected as:

Córdoba-Jabonero, C., Sicard, M., López-Cayuela, M.-A., Ansmann, A., Comerón, A., Zorzano, M.-P., Rodríguez-Gómez, A., and Muñoz-Porcar, C.: Aerosol radiative effect during the summer 2019 heatwave produced partly by an intercontinental Saharan dust outbreak - Part 1: Shortwave dust-induced direct impact, Atmospheric Chemistry and Physics, 21, 1–25, https://doi.org/10.5194/acp-21-6455-2021, 2021

---

## Author Comment (AC2)

Dear reviewers and editor:

We would like to thank you for taking the time to review our manuscript and suggest such valuable information in order to improve its quality. Even the most trivial suggestion helped in the progress of the text's value.

Some main and relevant changes have been made to the manuscript, as summarized below:

- While submitting the first version of our manuscript, a relevant study related to Aeolus optical products were made. Thus, for the sake of completeness, the following published paper has been included now in the reviewed version of our work:

  Baars, H., Radenz, M., Floutsi, A. A., Engelmann, R., Althausen, D., Heese, B., Ansmann, A., Flament, T., Dabas, A., Trapon, D., Reitebuch, O., Bley, S., & Wandinger, U. (2021). Californian Wildfire Smoke Over Europe: A First Example of the Aerosol Observing Capabilities of Aeolus Compared to Ground-Based Lidar. Geophysical Research Letters, 48(8), e2020GL092194. https://doi.org/10.1029/2020GL092194

- Furthermore, a reprocessing of Granada's ground-based dataset was made right after the manuscript submission, to increase the quality of the depolarization-related products. No relevant differences were found between the results obtained with the former and the reprocessed products. However, the charts included in the manuscript have been updated accordingly. These changes affect Figures 5, 12, 13, 14 and 15 (a new figure is included following the Reviewer#2 suggestion, so these figures correspond now to Figure 5, 13, 14, 15, and 16, respectively).

Apart from these main changes, a set of minor changes and typos has been corrected throughout the document. Following this introduction, you may find a detailed answer to every Reviewer#2's comment. All of the listed changes and the ones suggested by the reviewers can be seen in the new version of the manuscript, marked in red.

**Reviewer 2**

1) *Introduction*

   *The introduction of the paper is lengthy and distracts from the actual content of the paper. I miss the point why of the descriptions of all the networks is there, while the paper is based on three ground based lidar stations on the Iberian peninsula. The fact that these stations are part of ACTRIS/EARLINET is relevant, especially for the quality control of the instruments as well as the central data processing that is harmonised in the network. I suggest that the introduction is substantially shortened and focuses on these main points. It is relevant to refer to the ground based intercomparisons/validation efforts of optical profiles from other active space borne sensors, as the applied methodology is largely taken from these previous efforts (e.g. colocation criteria and network design).*

*Similarly, the main purpose of the Aeolus mission is a technological demonstrator for active wind profile measurements from space, as well as a demonstrator for the impact of those data on operational numerical weather prediction of those space borne observations. The optical data are, from a point of view of the mission, a by-product. I suggest that the introduction is also shortened to help the reader.*

Accordingly to the reviewer#2's comment the introduction section has been substantially shortened, with approximately a 30% reduction of the section's text with respect to the previous version (apart from removing the corresponding references). Concretely, the following text has been removed from the manuscript's introduction:

- Aerosols are unevenly distributed, both horizontally and vertically, with significant concentrations over landforms such as deserts (e.g. Laurent et al., 2010; Heinold et al., 2011; Ansmann et al., 2011) or large populated urban areas (e.g. Landulfo et al., 2003; Sun et al., 2004). Aerosol particles frequently travel through the troposphere (e.g. Guerrero-Rascado et al., 2009; Preißler et al., 2011; Sicard et al., 2012a, 2012b; Pereira et al., 2014; Granados-Muñoz et al., 2016) and exceptionally at the lower stratosphere (e.g. Ansmann et al., 1997, 2018; Sawamura et al., 2012; Baars et al., 2019).

- Nowadays, Global Observing Systems (GOS) allow the study of a great variety of atmospheric properties through ground-based instruments and satellites. Ground-based instruments for aerosol monitoring are set at local stations, which are unequally distributed in space, grouped in federated networks such as AERONET (Holben et al., 1998), EARLINET (Pappalardo et al., 2014), LALINET (Guerrero-Rascado et al., 2016; Antuña-Marrero et al, 2017) and MPLNET (Welton et al., 2006), and also unequally distributed in time, i.e. during intensive field campaigns, such as ACE (Bernath et al., 2005), SAMUM (Heintzenberg, 2009), SAMUM-2 (Knippertz et al., 2011) and CHARMEX (Mallet et al., 2016), among others.

- Nonetheless, until 2018, there was not a single satellite mission aimed to retrieve worldwide, continuous wind measurements from space.

- named after the Keeper of the Winds in Greek Mythology

- At that time, it became the fifth satellite in space of the ESA's Living Planet programme, the first European satellite with a lidar onboard and the first space-borne Doppler lidar ever able to measure vertical wind profiles on a global basis.

- During the 2020 pandemic of COVID-19, continuous near-real-time worldwide measurements from the Aeolus mission served as a prominent remedy to the lack of wind measurements, especially in the high troposphere and lower stratosphere, caused by air traffic reduction (Ingleby, 2020).

- ESA encourages the participation of organizations around the world, such as EARLINET, LALINET, NOAA (National Oceanic and Atmospheric Administration of the United States of America) and several other nation-wide organizations in Europe, Asia and North America.

2) *Section 2.1, Line 134-135 "Currently, L2A products access is still limited until a more confident version of the data products is achieved." This is a rather important statement. Here more explanation is needed. Perhaps the authors consider that the value of their manuscript may be devalued because of this since the conclusion will change after a new version of the Aeolus processor is released, but for the reader it is important to know a bit more about this.*

Recently, a new processor of the products was implemented, baseline 12. From May 26 2021, Aeolus products are processed under baseline 12 and these products are openly published. Thus, the quoted sentence is no longer accurate and will be restated and completed as:

Recently, L2A products began to be produced under a new processor version (baseline 12) and are openly published along L2B and L2C products.

3) *Section 2.2 A table with the station properties would be very helpful for a clear overview for the reader and gives the opportunity to shorten the lengthy descriptive text.*

This suggestion is particularly welcomed, as it will help to both reduce the manuscript length and also clarify some of the main aspects and differences of the ground stations. Thus, the following table has been included in the text:

| Station | | Granada (MULHACEN) | Évora (PAOLI) | Barcelona |
|---|---|---|---|---|
| Type | | Raman, elastic and depolarization | Raman, elastic and depolarization | Raman, elastic and depolarization |
| Laser radiation source | | Nd:YAG | Nd:YAG | Nd:YAG |
| Wavelengths (nm) | elastic | 355, 532, 1064 | 355, 532, 1064 | 355, 532, 1064 |
| | Raman | 354 ($N_2$), 407 ($H_2O$), 530 ($N_2$) | 387 ($N_2$), 607 ($N_2$) | 354 ($N_2$), 407 ($H_2O$), 607 ($N_2$) |
| | depol. | 532 | 532 | 355, 532 |
| Repetition frequency (Hz) | | 10 | 20 | 20 |
| Nominal vertical resolution (m) | | 7.5 | 30 | 3.75 |
| Nominal temporal resolution (s) | | 60 | 30 | 60 |
| Full overlap height (m agl) | | ~ 800 | ~ 800 | ~ 400 |
| References | | Guerrero-Rascado et al. (2010) Navas-Guzmán et al. (2011) Bravo-Aranda et al. (2013) | Preißler et al. (2011) | Kumar et al. (2011) Rodríguez-Gómez et al. (2017) Zenteno-Hernández et al. (2021) |

Table 1. Overview of the lidar systems of Granada, Évora and Barcelona stations.

4) *Section 3.1, Line 235 and further "In the current study, only aerosol products (L2A) are considered, and in particular particle backscatter coefficients derived from the Standard Correct Algorithm (SCA) and Standard Correct Algorithm middle bin (SCAmb)." Here the authors should explain why they limit themselves to the backscatter profiles and do not take into account the ALADIN L2A extinction profiles. After all, this is a first for space borne observations (CATS was configured to provide HSRL extinction profiles, but the instrument failed partially on this point.)*

The exploration of Aeolus L2A extinction profiles would have been a valuable contribution to this study. However, whereas backscatter profiles can be retrieved with high quality both during day and night-time, the retrieval of extinction profiles, with enough SNR is limited before sunrise and after sunset, when the Raman channels are not saturated by solar radiation. Depending on the time of the year, Aeolus overpasses take place almost around sunrise or sunset, when the background signal is highly variable, preventing its use for 1-hour (or 1.5-hour) averaged retrievals. At other times of the year Aeolus

overpasses take place way after sunrise or before sunset, so again no possible intercomparison of Aeolus extinction products can be assessed. Thus, the following information is now included in the manuscript:

The Raman derived extinction profiles retrieved at ground level could not be used in the study due to the time of the satellite overpass, when the signal-to-noise ratio is not good enough for these channels. Therefore, Aeolus extinction coefficients are not exploited in the study.

5) *Section 4.2.1, Line 365 A reference is made to profiles that are not shown. Please show the data.*

The following plot corresponds to the backscatter-related Ångström exponent profile calculated with the 355 and 532 nm channels of the measurements taken by the lidar system in Granada around the Aeolus overpass on 5th September 2019. We decided not to include this graphic in the manuscript in order to avoid extending the manuscript length. Furthermore, although the information of the graphic is relevant it is commented on the text with no need of including the plot, which might distract from the main objective of this section.

[Figure]

**Figure.** Backscatter-related Ångström exponent profile calculated at 355-532 nm ($\beta$-AE$_{355\text{-}532}$) derived by the lidar system in Granada around the Aeolus overpass of the 5th September 2019.

6) *Section 4.2.2, Line 391 A reference is made to sunphotometer measurements that are not shown. Please show the data.*

Following the reviewer#2's suggestion, the requested figure (Figure 8 in the text) is now included in the manuscript.

[Figure]

**Figure 8. Sun-photometer data retrieved by AERONET at Évora on the 28th June 2019: (a) AOD$_{675}$ and AOD-AE$_{440-870}$ and (b) fine mode fraction at 500 nm (FMF$_{500}$) and multiwavelength SSA daily series. The black vertical line indicates the Aeolus overpass.**

7) *Section 4.2.1, Line 408 A reference is made to profiles that are not shown. Please show the data.*

The following plot corresponds to the backscatter-related Ångström exponent profile calculated with the 355 and 532 nm channels of the measurements taken by the lidar system in Évora around the Aeolus overpass of the 28th June 2019. We decided not to include this graphic in the manuscript in order to avoid extending the manuscript length. Furthermore, although the information of the graphic is relevant it can be commented on the text with no need of including the plot, which might distract from the main objective of this section.

[Figure]

**Figure.** Backscatter-related Ångström exponent profile calculated at 355-532 nm ($\beta$-AE$_{355\text{-}532}$) derived by the lidar system in Évora around the Aeolus overpass of the 28th June 2019.

8) *Section 4.2.3 The case is presented as a smoke case, but proceeds to explain that there was a mixture of smoke and mineral dust. Please change the title of the section to remove the contradiction.*

We appreciate this suggestion. Thus, we will modify the title of Section 4.2.3 so that it reads:

**4.2.3 Case study of smoke and mineral dust mixture: Barcelona, 2nd July 2019**

9) *Section 5, Conclusions The first paragraph contains important information about the version of the data considered for the intercomparison. This should be mentioned either in the introduction or section 2.1 about Aeolus.*

The mentioned information appears in Section 3.1, where the processing of Aeolus products and baselines are mentioned and explained. The information regarding Aeolus included in the introduction or Section 2.1 is rather more general or more technical. However, a short introduction is now included in Section 2.1. Thus, the following information will be included in the text:

At the time of writing this article, the longest, fully homogeneous product dataset has been reprocessed in baseline 10 (B10). In this study, we evaluated Aeolus B10 optical products with a thorough analysis of Aeolus co-polar backscatter coefficients under the SCA and the SCAmb.

---

## Author Response (AR2)

Dear reviewers and editor:

We would like to thank you for taking the time to review our manuscript and suggest such valuable information in order to improve its quality. Even the most trivial suggestion helped in the progress of the text's value.

A set of minor changes and typos has been corrected throughout the document. Following this introduction, you may find a detailed answer to every Reviewer#1's comment. All of the listed changes and the ones suggested by the reviewers can be seen in the new version of the manuscript, marked in red. Additionally, a few more references have been added to the manuscript.

**Reviewer 1**

1) *There are many ESA internal references which seem to be not publicly available. All reference must be available somehow. You might discuss with ESA to submit these documents as supplement or ask if they can make it publicly available. I just googled some of them, an many documents can be downloaded, like for example the "Aeolus-Sensor-and-Product-Description ", which is freely available here:*
   *https://earth.esa.int/eogateway/documents/20142/37627/Aeolus-Sensor-and-Product-Description.pdf*
   *Please add links to all ESA documents and or make them available to the reader in another way.*

Following reviewer#1's request, the following links have been added to the manuscript's references section:
- European Space Agency, ESA: ADM-Aeolus Science Report, ESA SP-1311, https://earth.esa.int/pi/esa?id=3409&sideExpandedNavigationBoxId=Aos&cmd=image&topSelectedNavigationNodeId=AOS&targetIFramePage=/web/guest/pi-community/apply-for-data/ao-s&ts=1496439496255&type=file&colorTheme=03&sideNavigationType=AO&table=aotarget (last accessed on 21/11/2021) 2008.
- Flamant, P. H., Lever, V., Martinet, P., Flament, T., Cuesta, J., Dabas, A., Olivier, M., and Huber, D.: ADM-Aeolus L2A Algorithm Theoretical Baseline Document Particle spin-off products, ESA, reference: AE-TN-IPSL-GS-001, https://earth.esa.int/eogateway/documents/20142/37627/Aeolus-L2A-Algorithm-Theoretical-Baseline-Document (last accessed on 21/11/2021), 2020.
- Ingmann, P., and Straume, A. G.: ADM-Aeolus Mission Requirements Document, ESA, reference: AE-RP-ESA-SY-001 EOP-SM/2047, https://earth.esa.int/eogateway/documents/20142/1564626/Aeolus-Mission-Requirements.pdf (last accessed on 21/11/2021), 2016.
- Reitebuch, O., Huber D., and Nikolaus, I.: ADM-Aeolus Algorithm Theoretical Basis Document ATBD Level 1B Products, https://earth.esa.int/eogateway/documents/20142/37627/Aeolus-L1B-Algorithm-ATBD.pdf (last accessed on 21/11/2021), 2018.

- Straume, A. G., Schuettemeyer, D., Von Bismarck, J., Kanitz, T., and Fehr, T.: Aeolus Scientific Calibration and Validation Implementation Plan, https://earth.esa.int/eogateway/documents/20142/1564626/Aeolus-Scientific-CAL-VAL-Implementation-Plan.pdf (last accessed on 21/11/2021), 2019.

2) *With respect to the topic above, would like to mention that a description of the L2A retrievals is currently in discussion in the same special issue. Even though not yet accepted, it might be worth to cite:*
*https://amt.copernicus.org/preprints/amt-2021-181/*

Following reviewer#1's comment, the mentioned reference has been added to and cited in the manuscript.

3) *I am still struggling with the quantity you named BIAS. For me, a bias is a systematic error. But what you calculate is the Difference between ground-based lidar and Aeolus for some cases. As the reason of these differences could be also due to atmospheric nature (horizontal heterogeneity etc), I would prefer not to use this wording. Thus, I recommend to rename this parameter simply to (absolute) difference or something similar, but not bias, please.*

As the reviewer#1 points out, the use of bias for the quantity computed in this work is not rigorous. Strictly speaking, the term bias of an estimator in statistics refers to the difference between the mean value and the numerical value of the parameter it estimates. Therefore, we agree with the reviewer that the use of bias in the previous version of the manuscript is somewhat ambiguous leading to a misunderstanding, and therefore we propose to replace it by 'difference ($\Delta$)' through the manuscript.

4) *In my opinion, Figure 15 is not useful. It is known, that Aeolus uncertainty estimates (even though not perfect yet) handle only statistical uncertainties. Thus, it is not meaningful to compare them to the absolute differences (BIAS) derived from the comparison.*

Following reviewer#1 comment, former Figure 15 and the associated discussion has been removed from the manuscript.

5) *Reviewer 1, comment 2:*
*R1: "…It is not very clear why the authors prefer to use the literature values instead of spectral conversion factor retrieved from measurements."*
*Authors: "…Considering all these arguments, we prefered to use the spectral conversion factor derived from the literature in order to base our results on measurements of different stations (i.e. different environmental conditions and aerosol types), which themselves have been reviewed through peer review process."*
*Me: I understand your arguments and can agree on your conclusions, but why to show "your own" conversion factors then, if they are not used? This is confusing and has nothing to do with the topic! Please always have in mind: "Illustrations should only be shown if they are necessary for the understanding of the paper, not because they have been created. " Therefore, I recommend to remove Fig. 2a and b, and only show Fig. 2c. The comparison of different depolarization ratio at different wavelength is not the topic of your paper but could be followed in another study.*

Following this reviewer#1 comment and in order to make Section 4.1 easier to understand, we have decided to remove Figures 2b and 2c. We want to make it clear that we have used the literature-derived spectral conversion factor $K_\delta = 0.82 \pm 0.02$ from Figure 2a (now renamed as Figure 2).

6) *Reviewer 1, comment 4:*
   *"In section 4.3 Case studies, the authors stated "Sun -photometer measurements are taken into account for the sake of completeness aerosol typing, through the study of the aerosol optical depth at 675 nm (AOD 675 )". Why was the 657 nm wavelength chosen? Why not choose the AOD values in the UV region as 340 or 380 nm, instead?"*
   *Me: In my opinion your response is not sufficient. As reviewer 1 noted, you could have used a wavelength much closer to Aeolus (440 nm). And I don't think that 675 nm is most used in the community and is not the closed one to the wavelengths of your lidars. So I also do not see the point why to use that wavelength. Anyhow, as the AOD is only of minor importance for your work, it is not crucial, but you might consider to change this.*

Following reviewer#1's suggestion, we have replaced AOD675 values with AOD440 ones in the discussion of the Sun-photometer measurements, which is closer to the Aeolus wavelength and still with low uncertainty (~0.01) (Eck et al., 1999) .

7) *Reviewer 1, comments 5:*
   *Authors:" Regarding Barcelona station location just in the coastline, as the satellite overpasses the station at a close distance we can assume that both instruments (ground-based and space-borne lidars) detect the same air masses. Consequently, both instruments will register the same effects that the geographical layout might produce, so no special considerations have to be taken into account in the statistical analysis."*
   *Again I think your response is not sufficient. In my opinion you cannot assume that "both instruments detect the same airmasses" given the complex location of Barcelona and the very coarse resolution of Aeolus with 8 km in the horizontal. Rather, I would assume, that the coarse horizontal resolution of Aeolus and the horizontal heterogeneity in this area is one of the main reasons for the discrepancies found. Therefore, I think, you should discuss this issue in the paper with some sentences. Then, your newly added sentences in the conclusion would also fit much better, as it holds not only for the vertical but also for the horizontal domain.*

We agree with reviewer#1 and therefore we have decided to include the following sentence at the end of the Conclusions section:

"However, as it can be noted from the results, Aeolus vertical and horizontal resolution is too coarse (especially compared to other satellites) for a detailed characterization of the nuances of the atmospheric optical properties, especially in regions such as Barcelona, where the coarse horizontal resolution might cover very different orographic features, from a mountainous system to the Mediterranean sea. Thus, Aeolus provides valuable information in the detection and characterization of significant aerosol and cloud layers".

8) *Reviewer 1, comment 9:*
   *"9) Page 15 - lines 474-475 - "With the implementation of the quality flags (Figure 12c and 12 d), all of the sets range from 0 Mm-1sr -1 .". Please, consider correct this sentence. All the sets range from 0 to which value?*
   *Authors: The quoted sentence has been restated and completed as: With the implementation of the quality flags (Figure 13c and 13d), all of the sets range from 0 Mm-1sr-1 onwards. Actually, the maximum values mentioned are still flagged as valid, 86 Mm-1 sr-1 and 68 Mm-1 sr-1 in the case of the SCA and SCAmb, respectively."*

> *Me: the sentence is still awkward. I guess what you simply mean is that with the implementation of the QA/QC flags, no negative values are existent anymore?*

Following this reviewer#1 comment the mentioned sentence has been stated as:

"With the implementation of the quality flags (Figures 13c and 13d) any of the sets present negative backscatter coefficients".

> 9) *Reviewer 1, comment 10:*
> *Authors: "Aeolus backscatter coefficient uncertainties (known as Aeolus error estimates) are addressed through the biases between satellite and ground-based measurements (as presented in Section 3.3)."*
> *It is known, that the Aeolus uncertainties (even though not full developed) handle only statistical errors. So how can this be addressed through "your" BIAS?--> makes Fig. 15 obsolete.*

We agree with this reviewer#1 comment. Because of the current status of the Aeolus uncertainties representing only the statistical errors, Figure 15 has been removed in the new version of the manuscript.

> 10) *Line 428: Why are values up to 85 Mm^-1 sr^-1 unrealistic? Please explain or give reference!*

Following this reviewer#1's comment we have included some references to indicate that a backscatter coefficient of 86 Mm$^{-1}$ sr$^{-1}$ can be considered unrealistic. Consequently, the next sentences has been included in the manuscript:

"The mentioned large values (86 Mm$^{-1}$ sr$^{-1}$ in the case of the SCA and 68 Mm$^{-1}$ sr$^{-1}$ in the case of the SCAmb) can be considered as unrealistic when compared to particle backscatter coefficients detected during extreme events in the Iberian Peninsula (e.g. Guerrero-Rascado et al., 2009; Preißler et al., 2011; Cazorla et al., 2017; Fernández et al., 2019)".

> 11) *Line558: I would use "Nevertheless" or "Even though" instead of "Thus"*

Done.

> 12) *Line 935: Is the WMO report available somewhere? DOI? Internet page?*

Following this comment of reviewer#1, the web link of the mentioned report has been added to its reference:

"World Meteorological Organization, WMO: Proceedings of the third WMO Workshop on the impact of various observing systems on numerical weather prediction, WMO, https://library.wmo.int/doc_num.php?explnum_id=5409 (last access on 21/11/2021), 2004".

> 13) *Table 2 caption: Please write at least in the caption the full name of delta_par_linear. For example: "The linear particle depolarization ratio delta_par_linear at 355 nm…." And please indicate the UNIT, so that one at least knows that the unit is not [%].*

According to this comment from reviewer#1, the full name and the units of the variable has been written in the caption of Table 2 as follows:

"Table 2. Linear particle depolarization ratios ($\delta_{linear}^{part}$) at 355 and 532 nm (with the corresponding standard deviation) obtained from the literature for dust, marine and mixed anthropogenic aerosol types. It should be notice that $\delta_{linear}^{part}$ is a dimensionless variable".

*14) Fig. 1, Caption: in spring 2021, the orbit has changed, can you indicate that here: E.g. "Distribution of Aeolus overpasses during the studying period from .. too..."*

Following this reviewer#1 request we have specified the considered period in Figure 1 as:

"Figure 1. (a) Distribution of Aeolus overpasses over Europe during the considered period (B10, from July 2019 to December 2019 and from 20th April 2020 to 6th October 2020). (b) Location of the stations in Évora, Granada and Barcelona and the associated overpasses during the case studies analyzed in Section 4.2.. Source: ESA Aeolus online dissemination (aeolus-ds.eo.esa.int)."

*15) Figure 2b Caption: Please indicate were the measurements were taken.*

According to this reviewer#1 comment the region where the measurements were taken is now included in Table 2.

*16) Figure 3. Caption: Please remove "Daily"*

Following reviewer#1 comment, the word "daily" has been omitted from the caption of Figure 3, and for consistency also from Figures 6 and 9 captions.

*17) Fig 13 caption: Caption does not describe what is seen. Please write: Frequency disruption plot of ... derived from.... Also, please make the caption more concise.*

This reviewer#1 suggestion is taken into account and the caption of Figure 13 has been restated as:

"Figure 13. Histograms for the dataset, considering the three stations, of (a) Aeolus SCA co-polar backscatter coefficient without the implementation of quality flags; (b) Aeolus SCAmb co-polar backscatter coefficient without the implementation of quality flags. (c) is the same as (a) but considering quality flags. (d) is the same as (b) but considering quality flags".

*18) Fig. 14, caption: What is meant with "combined database"? Please write more exactly.*

This reviewer#1 suggestion is taken into account and the caption of Figure 14 has been restated as:

"Figure 14. (a) $\beta^{part}_{Aeolus\ like,355}$ and $\beta_{Aeolus\ SCA}$ of the dataset, considering the three stations, with no quality flags applied. (b) $\beta^{part}_{Aeolus\ like,355}$ and $\beta_{Aeolus\ SCAmb}$ of the dataset, considering the three stations, with no quality flags applied. (c) is the same as (a) but considering quality flags. (d) is the same as (b) but considering quality flags. The values of each dataset have been adjusted to a linear model with null intercept (red line)".

*19) Figure 15: See comment on Bias and Aeolus uncertainty above. In my opinion these plots are not useful.*

Following this comment and previous comments of reviewer#1, former Figure 15 and its related discussion has been omitted from the manuscript.

*20) Caption Fig. 16: Replace measures by measurements.*

Done. For the sake of consistency the same change has been made to Figures 15, 16 and 17 (former Figures 16, 17 and 18).

References

Eck, T. F., Holben, B. N., Reid, J. S., Dubovik, O., Smirnov, A., O'Neill, N. T., Slutsker, I., and Kinne, S.: Wavelength dependence of the optical depth of biomass burning, urban, and desert dust aerosols, J. Geophys. Res., 104( D24), 31333– 31349, https://doi.org/10.1029/1999JD900923, 1999.